 **eLIFE**

# Immediate perception of a reward is distinct from the reward's long-term salience

**John P McGinnis[1,2], Huoqing Jiang[1], Moutaz Ali Agha[3,4], Consuelo Perez Sanchez[1], Jeff Lange[1], Zulin Yu[1], Frederic Marion-Poll[3,4], Kausik Si[1,5]***

[1]Stowers Institute for Medical Research, Kansas City, United States; [2]Department of Molecular and Integrative Physiology, University of Kansas Medical Center, Kansas City, United States; [3]Evolution, Génomes, Comportement & Ecologie, CNRS, IRD, Université Paris-Sud, Université Paris-Saclay, Paris, France; [4]AgroParisTech, Paris, France; [5]Department of Integrative and Molecular Physiology, University of Kansas School of Medicine, Kansas City, United States

**Abstract** Reward perception guides all aspects of animal behavior. However, the relationship between the perceived value of a reward, the latent value of a reward, and the behavioral response remains unclear. Here we report that, given a choice between two sweet and chemically similar sugars—L- and D-arabinose—*Drosophila melanogaster* prefers D- over L- arabinose, but forms long-term memories of L-arabinose more reliably. Behavioral assays indicate that L-arabinose-generated memories require sugar receptor Gr43a, and calcium imaging and electrophysiological recordings indicate that L- and D-arabinose differentially activate Gr43a-expressing neurons. We posit that the immediate valence of a reward is not always predictive of the long-term reinforcement value of that reward, and that a subset of sugar-sensing neurons may generate distinct representations of similar sugars, allowing for rapid assessment of the salient features of various sugar rewards and generation of reward-specific behaviors. However, how sensory neurons communicate information about L-arabinose quality and concentration—features relevant for long-term memory—remains unknown.

*For correspondence: ksi@stowers.org

**Competing interests:** The authors declare that no competing interests exist.

## Introduction

In an environment filled with various stimuli, the positive experiences an animal remembers are widely assumed to be rewarding and salient. Long-term associative memories in particular are supposed to reflect the intensity of past responses to rewards. The experiences we remember, however, are not always those we expect to remember. How immediate reward perceptions influence future actions is therefore of wide interest.

Among various positive rewards, food, and in particular sweet food, has been most revealing since it is a source of both pleasure (immediate value) and nutrition (long-term value). Food is also a complex reward. Having evolved in distinct ecological niches, different species of *Drosophila* display distinct food preferences and discriminate between potential sources of nutrition (*Dethier, 1976*). For example, while some species of *Drosophila* prefer rotting fruits, others prefer mushrooms, cacti, or hibiscus flowers (*Markow and O'Grady, 2005*). Identifying and remembering relevant food, therefore, is essential for survival. Moreover, food is often not a single substance but a mixture of various compounds, and not all are equally rewarding: rotting fruits contain various sugars, alcohols, and acids that produce varying responses (*Yarmolinsky et al., 2009*; *Charlu et al., 2013*). Food in

**eLife digest** We often remember experiences that are rewarding in some way. However, not every rewarding experience is stored in memory, and the particular experiences we remember are not always those we would expect to remember. Why is it that some experiences generate long-term memories whereas others do not?

Fruit flies feed on a variety of different sugars present in rotting fruits. Although the flies find all of these sugars attractive, they form memories of some sugars more readily than others. This distinction is particularly striking in the case of two sugars with similar structures: D-arabinose and L-arabinose. Flies typically prefer D-arabinose over L-arabinose, but are more likely to remember an encounter with L-arabinose than D-arabinose.

McGinnis et al. have used fruit flies to explore how the rewarding properties of an experience affect how likely it is to be stored in memory. The experiments show that D-arabinose and L-arabinose generate different patterns of activity in the fly brain, and identify a subset of taste neurons that support the formation of memories specifically about L-arabinose. These neurons enable flies to associate features of their environment – such as odors – with the presence of this one particular sugar. Such memories may help the flies to find a similar food source again in the future. Artificially activating these neurons is also sufficient to trigger the formation of a memory, even in the absence of L-arabinose itself.

Taken as a whole, this work demonstrates that the immediate appeal of a reward can be separated from its ability to generate a long-term memory. The fact that activation of taste neurons can trigger memory formation explains how flies can quickly form long-term memories about desirable food sources. Looking ahead, further work will be required to understand the mechanisms that determine what animals like at any given moment, and what they remember over time.

natural contexts is also always part of an environment filled with other features, including predators, and therefore quick evaluation of potential food sources requires simultaneous processing of multiple stimuli. Finally, the attraction to food, and memories of it, are influenced by the internal state of the organism, such as whether the animal is hungry or satiated (*Colomb et al., 2009*; *Krashes et al., 2009*; *Toshima and Tanimura, 2012*; *Dethier, 1976*). It is therefore likely that contingent on their internal state, animals use certain components of food sources to quickly recognize those that are appropriate for feeding and, if worthwhile, to form memories of these sources for future visits. How these different aspects of food very quickly generate appropriate memories that guide future food-seeking behavior, however, remains unclear.

One possibility is that whatever components of food are most salient for long-term behavior are the same features that animals find immediately rewarding. This would predict that the more appealing (or palatable) a sugar is, the better it will be remembered. Another possibility is that certain components of food can reinforce memory relatively independent of the food's immediate appeal, because they indicate specific attributes of the food (e.g. nutritional content) that are of long-term relevance. In a complex environment, where an animal needs to process multiple stimuli simultaneously, such processing may ensure that regardless of the immediate response, stimuli of long-term relevance will be remembered.

In the course of exploring both the immediate appeal of various natural sugars and their ability to generate long-term associative memories, we serendipitously discovered that these two processes are separable. A specific illustration of this phenomenon is seen with the two chemically similar sugars, D- and L-arabinose: flies greatly prefer D-arabinose to L-arabinose, but better remember an odor paired with L-arabinose than with D-arabinose. We have also begun to explore how an animal assesses whether an experience that is rewarding in the moment is also of long-term relevance. Many studies have characterized higher order systems, particularly the neuromodulatory systems such as dopaminergic (*Schwaerzel et al., 2003*; *Huetteroth et al., 2015*; *Berry et al., 2012*; *Liu et al., 2012*; *Yamagata et al., 2015*; *Musso et al., 2015*), octopaminergic (*Burke et al., 2012*; *Schwaerzel et al., 2003*), neuropeptide F (*Krashes et al., 2009*) and mushroom body neurons (*Aso et al., 2014*; *Kirkhart and Scott, 2015*; *Vogt et al., 2014*) underlying long-term sugar reward

memory in *Drosophila*. How various sugars differentially engage the higher order reward system, however, remains unclear. We find that D- and L-arabinose differentially activate the same peripheral Gr43a-expressing neurons, and that activating Gr43a in some but not all manners can substitute for the sugar reward, indicating that sensory neurons can at least partially mediate this discrimination process. However, the exact mechanism by which these sensory neurons communicate the relevant features of L-arabinose to higher order systems remains unclear at this stage.

## Results

### *Drosophila melanogaster* prefers D-arabinose but more reliably forms long-term memories of odors paired with L-arabinose

To explore how animals evaluate salient features of food, we used an associative-appetitive memory paradigm (henceforth referred to as the 'memory paradigm') with *Drosophila melanogaster* that approximates food-seeking behavior (*Colomb et al., 2009*; *Krashes and Waddell, 2011*; *Tempel et al., 1983*). In this paradigm, hungry flies are trained for 2 min to associate an odor with a rewarding sweet sugar; trained flies subsequently seek out the sugar-associated odor for several days afterwards, indicating that they have formed an associative memory (*Figure 1A*). We have used this paradigm for three reasons: one, it is an ethologically relevant behavior; two, both the internal state (hunger) of the fly and characteristics of the sugar dictate the duration of memory (*Burke and Waddell, 2011*; *Fujita and Tanimura, 2011*; *Colomb et al., 2009*); and three, salient features of the sugar are evaluated rapidly within the 2-min training as reported by others (*Burke and Waddell, 2011*) and similarly confirmed by us (*Figure 1—figure supplement 1A and B*).

In the course of training flies (*Figure 1A*) with various sugars, including those that are present in *Drosophila melanogaster's* natural diet of ripening fruits (*Figure 1B*) some of which are non nutritious (*Figure 1C*), we observed that the relative appeal of a sugar (preference, *Figure 1D*) or short-term memory (minutes after training, *Figure 1E*) does not always predict its ability to act as a rewarding stimulus for long-term (24 hr after training) associative memory (*Figure 1F* and *Figure 1—figure supplement 1C*). This was apparent for multiple sugars, but nowhere so striking as the difference between two structural isomers, D- and L-arabinose (*Figure 2A*). D- and L-arabinose both taste sweet (*Figure 2—figure supplement 1*) and are both non-nutritious (*Figure 2B*). Flies overwhelmingly preferred D-arabinose to L- (*Figure 1D* and *Figure 1—figure supplement 1C*), and form similar short-term memories of both sugars (*Figure 1E* and *Figure 2C*). However, it is L-arabinose, not D-, that is more effective in producing long-term memory (*Figure 1F* and *Figure 2C*). The relative ineffectiveness of D-arabinose in producing long-term memory is consistent with other studies (*Burke and Waddell, 2011*; *Fujita and Tanimura, 2011*; *Cervantes-Sandoval and Davis, 2012*).

A trivial explanation for the observed memory with L-arabinose would be contamination with nutritious sugars. But L-arabinose bought from different sources generated similar survival curves and memory scores (*Figure 2—figure supplement 2A*). The L-arabinose memory is also not due to particular wild-type flies used in the experiment (*Figure 2—figure supplement 2B*), the training conditions or the particular experimenter (*Figure 2—figure supplement 2C*). Neither arabinogalactan (a polymer of L-arabinose and galactose) nor the natural L-sugar rhamnose, produce significant memory, indicating that not all L-arabinose-containing components of fruits' cell wall or natural L-sugars are conducive to memory formation (*Figure 2—figure supplement 2D*). Bacteria are known to utilize L-arabinose (*Watanabe et al., 2006*), but the flies' resident bacteria had no evident contribution to L-arabinose memory, since giving the flies a cocktail of antibiotics for the 2 days prior to behavioral training had no effect on L-arabinose memory (*Figure 2—figure supplement 2E*). Taken together, these results suggest that L-arabinose can act as a rewarding stimulus for long-term associative memory.

We wondered whether the behavioral differences between D- and L- were due to the high concentration (1 M) of sugars, although 1 M to 3M sugar is standard in memory assays (*Yamagata et al., 2015*; *Cervantes-Sandoval and Davis, 2012*; *Burke and Waddell, 2011*). However, the preference for D-arabinose (*Figure 2D*) persisted when the sugars' concentrations were both reduced 100-fold (10 mM), and began to shift only when the concentration of D-arabinose was reduced to less than a third of L-arabinose (*Figure 2—figure supplement 3A*). A similar difference between D- and L-arabinose has also been reported in the blowfly *Phormia regina*, where the taste

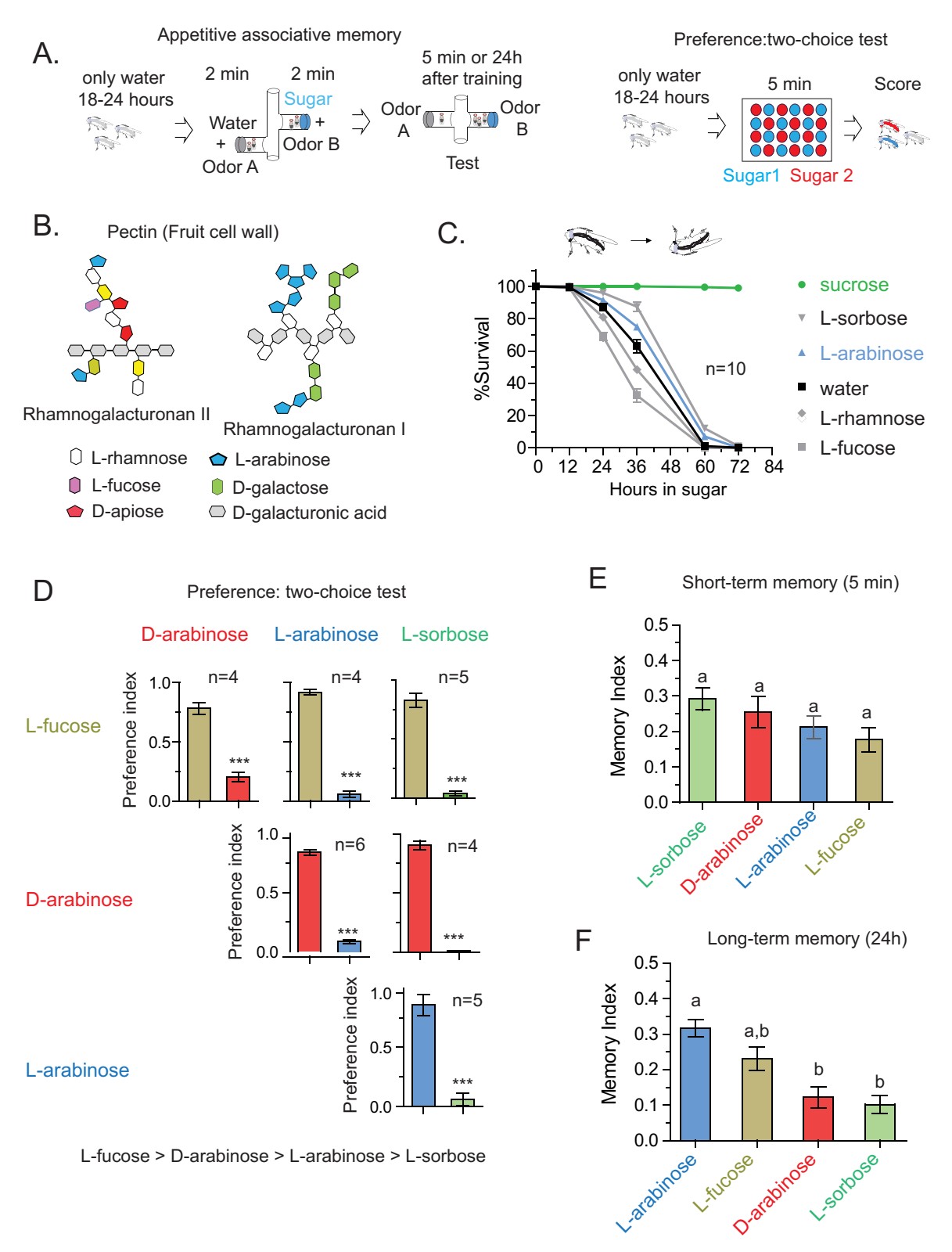

**Figure 1.** Flies' immediate preference for a sugar is not predictive of their long-term memory: various sugars from fruits. (**A**) Schematic of behavioral assays. In the appetitive associative memory paradigm, hungry flies are trained for 2 min with the sugar-odor pair and memory is assayed by subsequently giving a choice between the two odors. In the preference assay, hungry flies are given a choice between two sugars mixed with different colors; after 5 min color of the abdomen is used to assess consumption. (**B**) Schematic of selected pectic polysaccharides present in fruits' cell

*Figure 1 continued on next page*

*Figure 1 continued*

walls, adapted from Harholt et al. (**Harholt et al., 2010**). (C) Survival percentages for flies given solely 1 M sugar solutions. *n* = 10 (50 flies per *n*) for each time point. (D) Two-choice tests comparing flies' preference for each of four sugars when both sugars are presented side-by-side for 5 min (50 flies per n). (E) Short-term (5 min) associative memory scores for the sugars. (*n* = 7–11) (F) Long-term (24 hr) associative memory scores for the sugars. L-fucose is a component of pectin as well, although the amount is low compared to L-arabinose. (*n* = 20–24) Memory scores labeled *a* are significantly different (<0.05) from bars labeled *b*, analyzed by one way ANOVA with Tukey's multiple comparisons test. Detailed explanations of what constitutes a single *n* is found in Materials and methods. Results with error bars are means ± s.e.m.

The following figure supplement is available for figure 1:

**Figure supplement 1.** The CS-US association occurs during the two-minute training.

threshold for D-arabinose is reported to be five times lower than that of L-arabinose (**Hassett et al., 1950**). When we measured consumption by Capillary Feeder (CAFÉ) assays over a concentration range (**Figure 2—figure supplement 3B**) or by mixing radioactive [32]P in the food in fixed concentration (**Figure 2—figure supplement 3C**), the flies consumed more D- than L-arabinose. Over time, however, flies consumed less D-and L-arabinose than nutritious sugars (data not shown), consistent with other studies (**Dus et al., 2011**; **Stafford et al., 2012**), and consumption reached a plateau in ~14 min for D- and ~30 min for L. We also monitored by video the behavior of single flies as they fed on colorless D- and L-arabinose solutions (**Figure 2—figure supplement 3D**) and observed that they spend much more time on D-arabinose than L-arabinose, consistent with higher overall consumption. These differences are not due to differences in mere detection of D- and L- arabinose: detection rates were very similar at high concentrations and began to differ only when concentrations were dropped to ≤50 mM (**Figure 2—figure supplement 1**). Likewise, the ability of L-arabinose to generate long-lasting memory persisted even at a 10-fold lower concentration, albeit with much weaker efficacy (**Figure 2E**). Moreover, lowering D-arabinose concentration, where the flies still detect D-arabinose but consume less of it, there was no increase in memory (**Figure 2—figure supplement 3E**) ruling out the possibility that consuming too much non-nutritious sugar, such as D-arabinose, is somehow a negative reinforcement.

In addition to consumption, we also measured the proboscis extension response (PER), which reports immediate acceptance of a taste stimuli. Curiously, PER response was similar between D- and L-arabinose over a concentration range (**Figure 2—figure supplement 4A**), consistent with other reports that PER depends more on the intensity than chemical nature of the sugar (**Masek and Scott, 2010**; **Stafford et al., 2012**). However, mere detection and acceptance of the sugar is not sufficient for long-term memory: A choice test between water and various concentrations of sucrose (a potent inducer of long-term memory) showed that there was no difference in the likelihood of consumption between 1 M and 10 mM sucrose; only when sucrose concentration is reduced to 1 mM did detection begin to fall (**Figure 2—figure supplement 4B**). However, only sucrose concentrations ≥ 100 mM reliably produced robust long-term memory (**Figure 2—figure supplement 4C**). Therefore, various sugar-associated behavioral responses, such as detection, acceptance, and assessment of immediate and long-term relevance are not a single process and are likely dictated by various attributes of the sugar. Taken together, these results suggest that even two chemically similar sugars can elicit quite distinct short- and long-term behaviors, and that immediate behavioral responses are not always predictive of long-term behavioral consequences: while flies find D-arabinose more immediately appealing, L-arabinose is more salient for long-term memory.

## Gustatory neurons involved in L-arabinose memory and D-arabinose preference

What is the neural basis for the difference in behavioral responses to D- and L-arabinose? There are two possibilities, not mutually exclusive: the two sugars engage distinct neural pathways, or they activate the same neural pathways in a distinct manner. The intial step in sugar detection and consumption are the gustatory-receptor-expressing (Gr) neurons that respond to sweet substances. To date, Gr5a, Gr43a, Gr61a, and Gr64a, b, c, d, e, and f have been implicated in sweet sugar detection (**Dahanukar et al., 2007**; **Jiao et al., 2008**; **Yavuz et al., 2014**; **Freeman et al., 2014**; **Miyamoto et al., 2013**). We therefore used Gr-GAL4 drivers to express the inward rectifying

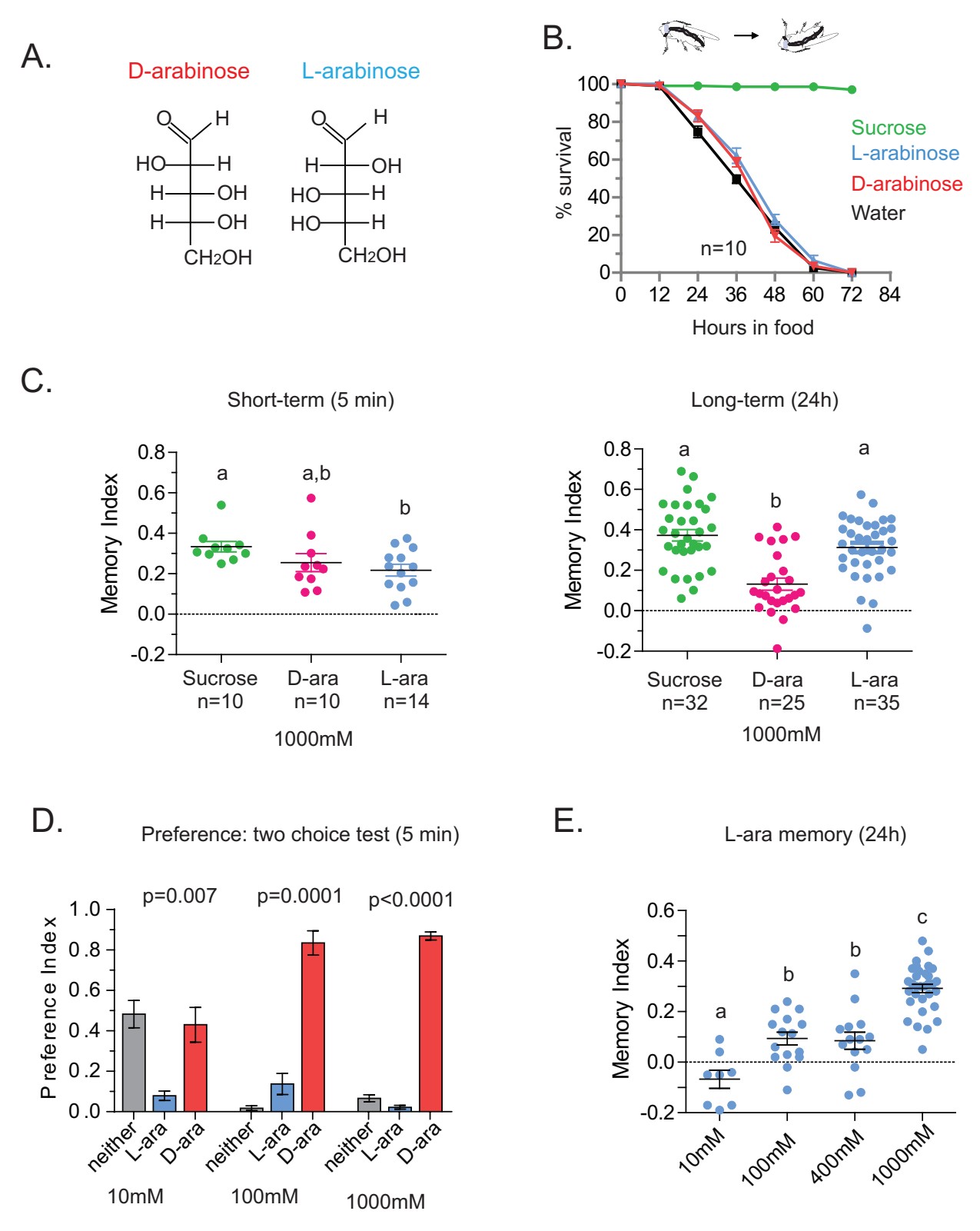

**Figure 2.** Flies' immediate preference for a sugar is not predictive of their long-term memory: L- vs D-arabinose. (**A**) Structures of D- and L-arabinose. (**B**) Survival percentages for flies given solely 1 M sugar solutions. $n = 10$ (50 flies per $n$) for each time point. (**C**) Short- and long-term memory of sucrose and D- and L-arabinose. (**D**) Two-choice tests comparing flies' preference for D- and L-arabinose when both sugars are presented side-by-side for 5 mins. $n = 4$ (50 flies per n). (**E**) Long-term (24 hr) memory scores for increasing concentrations of L-arabinose. Results with error bars are means ± s.e.m.
*Figure 2 continued on next page*

*Figure 2 continued*

ns, not significant. *$\leq$0.01, **$\leq$0.001 and ***$\leq$0.0001. The significant differences ($p < 0.05$) between conditions in *Figure 2C and E* were analyzed by one-way ANOVA with Tukey's multiple comparisons test and differences are denoted by different letters. Detailed explanations of what constitutes a single n is found in Materials and methods.

The following figure supplements are available for figure 2:

**Figure supplement 1.** Palatability of D- and L-arabinose over a concentration range.

**Figure supplement 2.** Specificity of L-arabinose memory.

**Figure supplement 3.** Although both are sweet, D-arabinose is preferred over L-arabinose.

**Figure supplement 4.** Detection and memory.

potassium channel Kir2.1 (Gr-GAL4/+; UAS-Kir2.1/+), silencing these sets of Gr-expressing neurons (*Baines et al., 2001*) in order to determine the neurons involved in D vs L preference and L-arabinose memory.

Preference and memory for a sugar starts with detecting the sugar; silencing neurons required for detection could cause a general decline in consumption of a particular sugar or all sugars. We therefore first measured the flies' ability to detect and consume L- or D-arabinose following silencing of specific Gr-expressing neurons (*Figure 3—figure supplement 1A and B*). Silencing Gr5a-expressing neurons reduced L-arabinose detection by about 80%, while silencing Gr61a-expressing neurons reduced both L-and D-arabinose detection by ~50%. Silencing Gr64e and Gr64f neurons almost completely abolished detection, discrimination and memory, consistent with previous reports that these receptors are likely expressed in all neurons responsible for sugar detection (*Jiao et al., 2008*; *Wisotsky et al., 2011*).

Silencing neurons required for discrimination would result in equivalent consumption of D- and L-arabinose. Upon silencing of Gr5a-, Gr43a-,Gr64a-, or Gr64d-expressing neurons, flies still overwhelmingly preferred D-arabinose (*Figure 3A*). Only silencing Gr61a neurons reduced D-arabinose consumption while increasing L-arabinose consumption (and ~30% flies did not eat any sugar), indicating that without Gr61a-expressing neurons flies were beginning to have trouble discriminating between the two sugars (*Figure 3A*).

In contrast to D-arabinose preference, silencing of Gr5a-, Gr43a-, Gr61a-, and Gr64f- but not Gr64a- or Gr64d-expressing neurons, significantly impaired L-arabinose memory (*Figure 3B*). Since silencing Gr5a- and Gr64f-expressing neurons also impairs L-arabinose consumption, the memory impairments may very well be due to an inability to detect L-arabinose (*Figure 3—figure supplement 1A*). Since silencing Gr61a neurons reduces detection, discrimination and memory, they may play a more general role in L- and D-arabinose detection and subsequent processing. Interestingly, silencing of Gr43a neurons had no effect on L-arabinose detection (*Figure 3—figure supplement 1A*) or D-arabinose preference (*Figure 3A*), but resulted in loss of L-arabinose memory (*Figure 3B*), suggesting Gr43a-expressing neurons play an important role in L-arabinose memory.

## Gr43a is required for L-arabinose memory

Single gustatory neurons express multiple gustatory receptors. To determine which receptor within Gr43a neurons—Gr43a or some other receptor expressed by these neurons—is important for L-arabinose memory we trained receptor mutants with L-arabinose. Δgr43a, Δgr61a, and Δgr43a-61a flies all showed a significant reduction ($p < 0.01$) in long-term memory at 24 hr (*Figure 3C*). However, D-arabinose preference is maintained in the absence of any single known sugar receptor (*Figure 3D*). To determine whether L-arabinose memory phenotypes were simply due to detection deficits, we tested the mutants' ability to detect L-arabinose. Deletion of Gr43a had a small effect on L-arabinose detection, and deletion of Gr61a resulted in ~40% reduction (*Figure 3—figure supplement 2A*). D-arabinose detection was not altered by any single receptor deletion (*Figure 3—figure supplement 2B*). Taken together, these results suggest that Gr43a and Gr43a-expressing neurons are important to form long-term memory of L-arabinose, while Gr61a and Gr61a-expressing

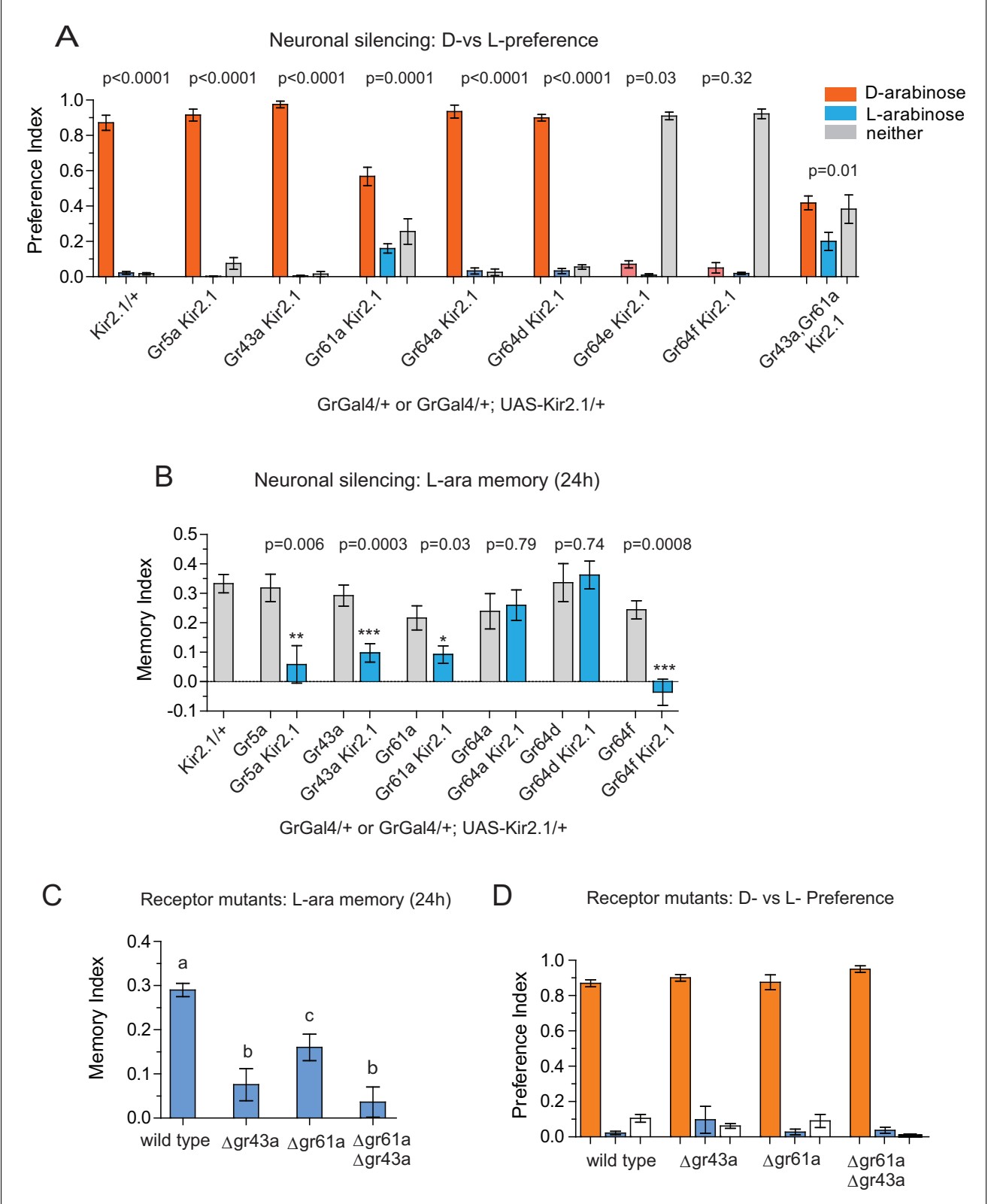

**Figure 3.** Gr43a- and Gr61a-expressing neurons are involved in D vs L preference and in L-arabinose memory. (**A**) Silencing of Gr61a-expressing neurons with Kir2.1 impaired D > L discrimination and preference; silencing Gr64e- and Gr64f-expressing neurons nearly eliminated detection of both sugars. (**B**) Silencing Gr43a- and Gr61a-expressing neurons impaired L-arabinose memory. Gr64aGAL4 and Gr64dGAL4, whose expression is restricted to LSO and VCSO neurons did not impair L-arabinose memory. Silencing Gr64f- and Gr5a-neurons reduce L-arabinose memory, but they also impair

*Figure 3 continued on next page*

*Figure 3 continued*

L-arabinose detection. (see *Figure 3—figure supplement 1*). (**C**) Gr43a and Gr61a receptors are important for L-arabinose memory. (**D**) No single receptor mutant impaired D > L preference. For multiple samples, one-way ANOVA with Tukey's multiple comparisons test was performed, and significant differences ($p<0.05$) are denoted by different letters. Results with error bars are means ± s.e.m. ns, not significant. *$\leq$0.01, **$\leq$0.001 and ***$\leq$0.0001.

The following figure supplements are available for figure 3:

**Figure supplement 1.** Gr expressing neurons involved in D- and L-arabinose detection.

**Figure supplement 2.** L- and D-arabinose detection do not rely on any single receptor.

neurons are important for L- and D-arabinose detection and discrimination. These results, however, do not rule out the possibility that there may be an unidentified receptor that exclusively mediates D-arabinose preference or that L-arabinose memory uses other receptors in addition to Gr43a and Gr61a.

## Peripheral Gr43a neurons are involved in L-arabinose memory

In *Drosophila*, gustatory receptors are present on the antennae, legs, wings, and labellae, and in the pharynx, gut, and central brain (*Joseph and Carlson, 2015*). The wide expression pattern, presence of multiple receptors in the same neurons, and different combinations of receptors in different neurons indicate that gustatory-receptor-expressing neurons in various locations may respond quite differently to different sugars (*Thoma et al., 2016*; *Miyamoto and Amrein, 2014*). We focused particularly on Gr43a-expressing neurons for their specific involvement in L-arabinose memory, and previous studies suggested they act as nutrient sensors (*Miyamoto et al., 2012*). We therefore sought to determine whether all Gr43a-expressing neurons or only a subset of Gr43a neurons are important for L-arabinose memory. As reported by others (*Miyamoto et al., 2012*; *Park and Kwon, 2011*), Gr43a$^{GAL4}$ expression is consistently detected in four dorsolateral protocerebrum (DLP) neurons in the central brain, the LSO and VCSO neurons in the proboscis, two f5 neurons in the distal tarsi, and in the proventricular ganglion of the gut (*Figure 4A*). We selectively silenced the central brain DLP neurons using a Gr43aGAL4:ChaGAL80 (*Miyamoto et al., 2012*) combination (*Figure 4B*) or the LSO and VCSO neurons using Gr64aGAL4 and Gr64dGAL4 (*Figure 3B*). While silencing all Gr43a neurons impaired L-arabinose memory, silencing of just the DLP (*Figure 4B*), or just the LSO and VCSO neurons (*Figure 3B*) had no significant effect, suggesting that some combination of Gr43a-expressing neurons that includes the tarsal and/or gut neurons are the necessary Gr43a-expressing neurons for L-arabinose memory. Because silencing of Gr61a- and Gr5a-expressing neurons each blocked L-arabinose memory (*Figure 3B*) and neither Gr5a nor Gr61a expression can be detected in the gut, it seems that the tarsal Gr43a-expressing neurons are the important ones for L-arabinose memory. Previous studies suggested that in the f5 neurons in the distal tarsi, Gr43a is coexpressed with Gr61a (*Figure 4—figure supplement 1A*) (*Freeman and Dahanukar, 2015*). There were uncertainities about the coexpression of Gr43a and Gr5a in distal tarsi. However, split-GAL4 reconstitution assay suggests that Gr5a and Gr43a are likely to be coexpressed in one f5 neuron (likely f5V) in the distal tarsi (*Figure 4—figure supplement 1B*), in agreement with previous work (*Miyamoto et al., 2012*). Taken together, these results suggest that f5 neurons in the distal tarsi coexpressing Gr43a and some combination of Gr5a and Gr61a are involved in L-arabinose memory. However, these results do not rule out the possibility that other Gr43a neurons or other Gr-expressing neurons are involved in L-arabinose memory.

## L-arabinose and D-arabinose activate peripheral Gr43a neurons to different extent

To understand how D- and L-arabinose generate different behavioral responses, we analyzed electrophysiological responses of f5V sensilla in the distal tarsi, which host neurons expressing Gr43a. D-arabinose consistently generated significantly more spikes over a range of concentrations (*Figure 5A and B*). Differences in electrophysiological response also manifested in calcium levels measured by GCaMP6, a genetically encoded calcium indicator (*Chen et al., 2013*). In Gr43a-

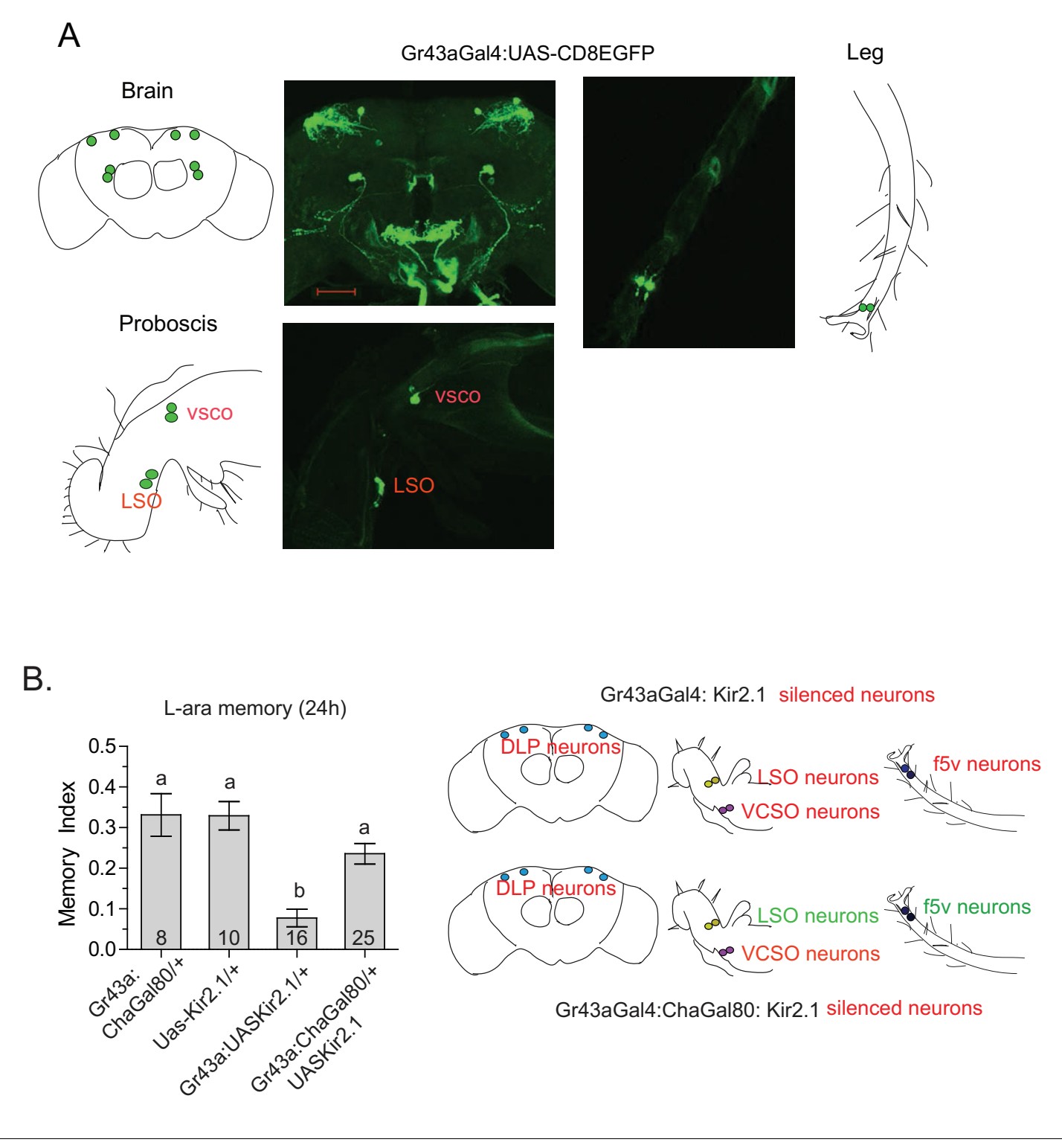

**Figure 4.** Tarsal Gr43a neurons are critical for L-arabinose memory. (**A**) Gr43a[GAL4] neurons are observed in the dorsolateral protocerebrum, central brain, proboscis, leg, and gut (not shown). (**B**) Silencing only the dorsal protocerebral (DLP) and VCSO neurons does not impair L-arabinose memory. Left panel: memory score in various genetic backgrounds. Right panel, top: in Gr43a[GAL4]/+; Kir2.1/+ flies, all indicated neurons are silenced (including proventricular neurons, not pictured). Bottom: in Gr43a[GAL4]:ChaGal80/+; UAS-Kir2.1/+ flies, only the neurons indicated in red type are silenced. For multiple samples, one-way ANOVA with Tukey's multiple comparisons test was performed, and significant differences (p<0.05) are denoted by different letters. Results with error bars are means ± s.e.m. ns, not significant. *≤0.01, **≤0.001 and ***≤0.0001.

*Figure 4 continued on next page*

*Figure 4 continued*

The following figure supplement is available for figure 4:

**Figure supplement 1.** Expression patterns of Gr-GAL4s.

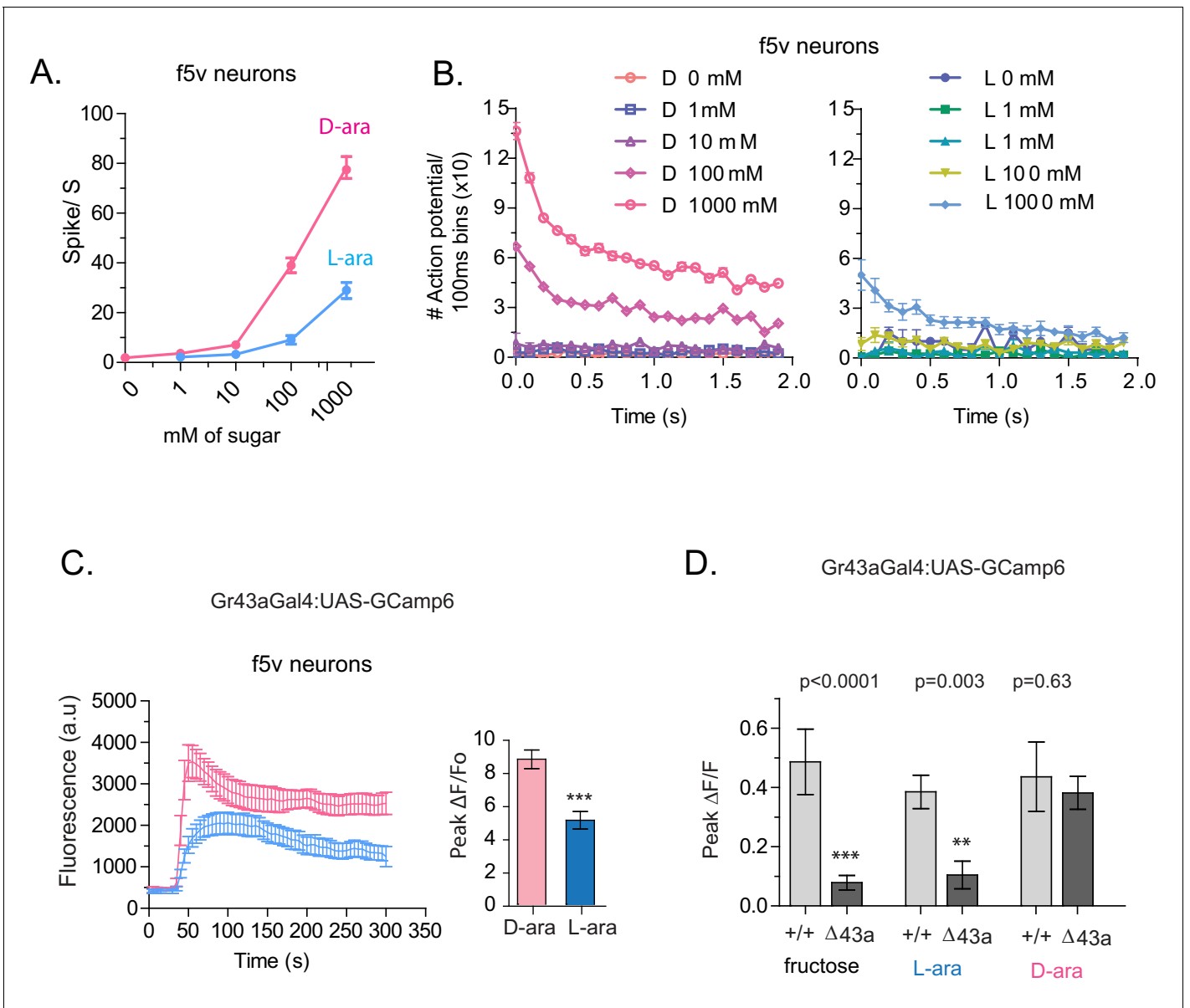

**Figure 5.** Tarsal Gr43a neurons respond differentially to D- and L-arabinose. (**A**) Spikes per second of recorded f5V tarsal neuron in response to D- or L-arabinose at various concentrations. (**B**) Spikes per second binned by 100 ms over the first two seconds of response. (**C**) The evoked-calcium activity of Gr43a[GAL4] neurons in the distal tarsi. (**D**) Removal of Gr43a receptor impairs fructose and L-arabinose activation of Gr43a[GAL4] neurons. Results with error bars are means ± s.e.m. ns, not significant. *$\leq$0.01, **$\leq$0.001 and ***$\leq$0.0001.

The following figure supplement is available for figure 5:

**Figure supplement 1.** Evoked-calcium activity of Gr43a neurons in response to D- and L-arabinose.

expressing f5 neurons in distal tarsi, the peak calcium level was higher and reached more rapidly for D-arabinose than L-arabinose (*Figure 5C*). Removal of the Gr43a receptor from these neurons significantly reduced response to L-arabinose but not D-arabinose (*Figure 5D*), consistent with the idea that D-arabinose activates multiple receptors. In the proboscis LSO neurons, there was a quicker rise and fall in response to L-arabinose with a slower but more sustained activation in response to D-arabinose (*Figure 5—figure supplement 1A*). However, such differences between D- and L-arabinose-provoked responses are not universal: the average D- and L- arabinose responses of the central brain DLP neurons were similar in both magnitude and shape (*Figure 5—figure supplement 1B*). These results indicate that D- and L-arabinose can activate the same gustatory neurons to different extents and that differential activation depends on properties specific to each neuron.

## Activation of Gr43a neurons can substitute the sugar reward to form associative memory

Because Gr43a$^{GAL4}$ offered the most restricted set of neurons that was critical for L-arabinose memory, we sought to determine whether they also represented the minimum set of gustatory neurons sufficient for appetitive long-term memory formation. To this end, we asked whether activating Gr43a$^{GAL4}$ neurons in the memory paradigm—in the absence of sugar—could generate an associative-appetitive memory (*Figure 6A*). dTrpA1, a temperature-sensitive cation channel, causes continuous activation of neurons at temperatures above 26°C (*Hamada et al., 2008*). Activation of Gr43a$^{GAL4}$ neurons by dTrpA1, however, failed to substitute for the sugar reward (*Figure 6B*), although similar activation of a subset of dopaminergic neurons (R58E02-GAL4/+; dTrpA1/+) produced long-term appetitive memory as reported by others (*Liu et al., 2012*) (*Figure 6—figure supplement 1A*). These results suggested either that activation of Gr43a$^{GAL4}$ neurons is necessary but not sufficient for L-arabinose memory, or that dTrpA1 does not approximate the activation required to produce long-term memory. Consistent with the latter possibility, activation of Gr43a neurons with the red-shifted channelrhodopsin variant ReaChR, a light-gated cation channel that depolarizes neurons in response to red light (*Lin et al., 2013*) produced associative memory: when flies expressing ReaChR in Gr43a$^{GAL4}$ neurons were exposed to one odor without the light, and a second odor in the presence of red light, the flies subsequently preferred the light-associated odor (*Figure 6C*). Intriguingly, activation by the same amount of light evenly distributed was not effective in producing long-term memory, suggesting that these patterns evoked different levels or patterns of activity in Gr43a neurons; the nature of this activation is unknown at this time (*Figure 6C*). Finally, starvation is an important regulator of memory strength in the associative-appetitive paradigm—the hungrier the flies are, the better memories they form (*Krashes et al., 2009*; *Colomb et al., 2009*). Starvation also influenced the memory strength following Gr43a-neuron activation: the same pulsated light activation produced memory in starved but not fed flies (*Figure 6D*).

Since the f5 neurons in the distal tarsi express Gr5a in addition to Gr43a, we also activated Gr5a-expressing neurons. Similar to Gr43a neurons, activation of Gr5a-expressing neurons resulted in robust long-term memory (*Figure 6E*). Activation of Gr64a-expressing neurons, which labels the LSO and VCSO neurons, did not produce significant long-term memory (*Figure 6E*). Interestingly, Gr61a-expressing neurons are necessary but not sufficient to generate associative memory (*Figure 6E*), suggesting that activation of some Gr43a and Gr5a expressing neurons could be critical for memory processes, or that activation of the additional Gr61a-expressing neurons somehow weakens the co-expressing neurons' likelihood of generating memory. Taken together, these results suggest that activation of a subset of Gr43a-expressing neurons is sufficient to generate long-lasting associative memory. These observations further suggest that activation of the same neurons by different methods, perhaps leading to different activity levels/patterns, give rise to substantially different behavioral outcomes, consistent with other reports (*Clark et al., 2013*; *Seeger-Armbruster et al., 2015*). However, further work is necessary to determine exactly which subsets of neurons contribute to L-arabinose memory, and whether these neurons needs to be activated in a specific pattern to elicit long-term memory.

## Discussion

The observation that two similar sugars generate strikingly different behavioral responses can perhaps be best understood using the framework of 'incentive salience' in rewards, formulated by

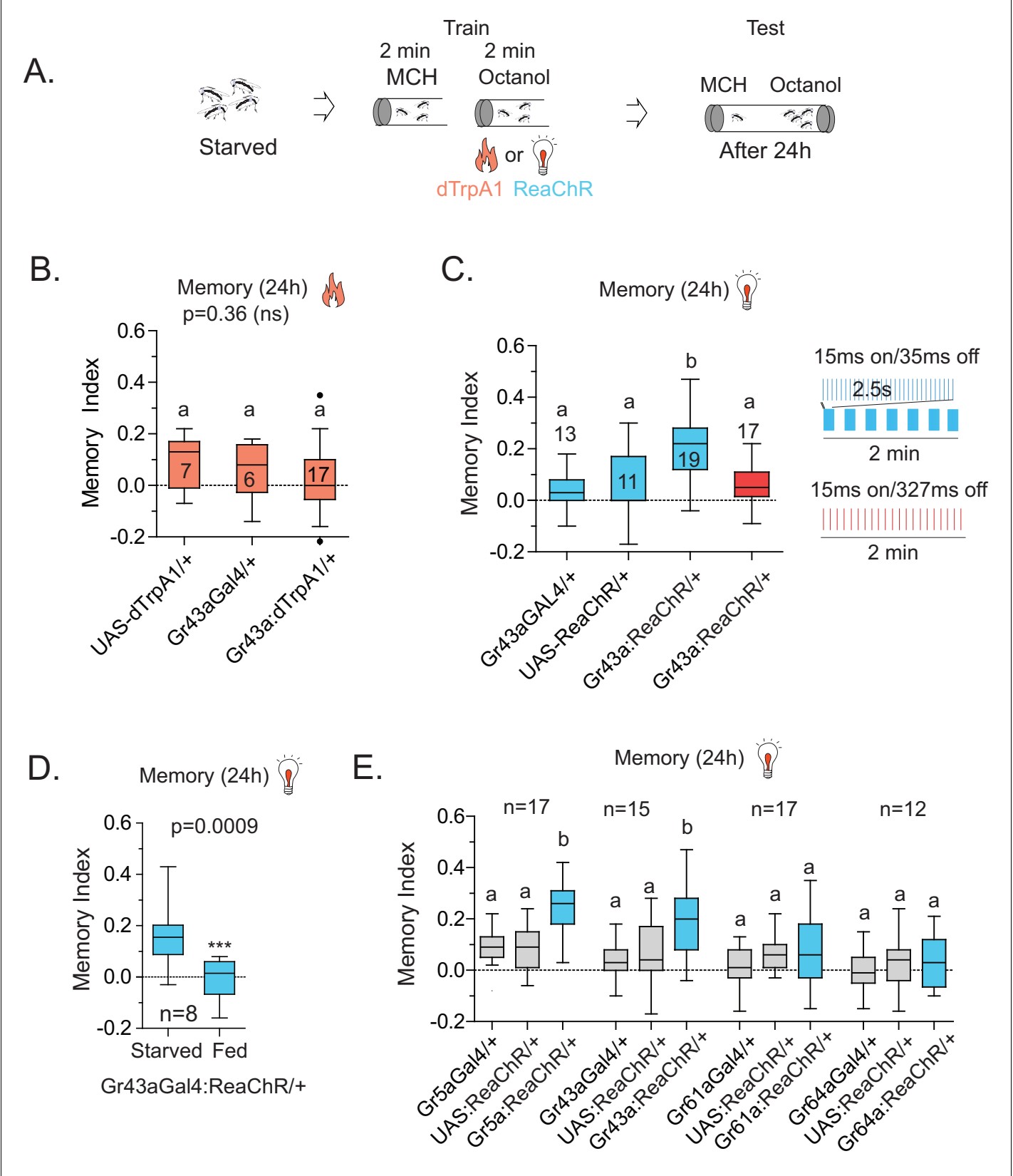

**Figure 6.** Activation of Gr43a neurons is sufficient to form rewarding associative memory. (**A**) Schematic of heat and light-activated associative olfactory training. (**B**) Activation of Gr43aGAL4 neurons by dTrpA1 (at 31 °C) does not induce long-term memory. (**C**) A 20 Hz, 15 ms pulse-width activation for 2.5

*Figure 6 continued on next page*

*Figure 6 continued*

s, repeated every 20 s, induces long-term memory in flies expressing ReaChR in Gr43a$^{GAL4}$ neurons; genetic controls do not show significant memory, and the same amount of light using the same pulse-width but distributed uniformly over the 2 min generates no memory (red). Schematics of light patterns are not to scale. (**D**) Optogenetic activation of Gr43a$^{GAL4}$ neurons induces memory only in hungry flies, not in flies fed ad libitum. (**E**) Optogenetic activation of Gr43a- and Gr5a-expressing neurons leads to substantial 24 hr memory; activation of Gr61a- or Gr64a-expressing neurons does not. For multiple samples, one-way ANOVA with Tukey's multiple comparisons test was performed, and significant differences (p<0.05) are denoted by different letters. Results with error bars are means ± s.e.m. ns, not significant. *$\leq$0.01, **$\leq$0.001 and ***$\leq$0.0001.

The following figure supplement is available for figure 6:

**Figure supplement 1.** dTrpA1 activation of R58E02 neurons does not produce long-term memory.

Berridge and Robinson (*Berridge and Robinson, 2003*), who divided reward percepts into 'liking' (conscious pleasure, hedonic) and 'wanting' (incentive salience). According to Berridge and Robinson, 'wanting' (incentive salience) is a component of rewards that transforms mere sensory information about rewards and their cues into 'attractive, desired, riveting incentives' and 'emerged early in evolution as an elementary form of stimulus-guided goal direction, to mediate pursuit of a few innate food or sex unconditioned stimuli' (*Berridge and Robinson, 2003*). In most cases, rewards that are 'liked' are usually also 'wanted', and in conventional formulations, they are considered effectively identical. But work on addiction and monetary reward on human suggest that 'wanting' and 'liking' are in fact dissociable, and while, in many cases, a behavioral response to an experience can predict the likelihood of memory formation, people can be motivated by cues remaining outside conscious awareness (*Pessiglione et al., 2007*; *Wise, 2002*). Here, we report that a similar distinction in reward perception may also exist in *Drosophila*, which is suggested by others (*Perry and Barron, 2013*): D-arabinose appears to preferentially involve the 'liking' component of the reward percept and L-arabinose the 'wanting'. For *Drosophila*, the incentive to remember L-arabinose is perhaps owing to the fact that it can inform a specific attribute of food, such as the ripening status of a fruit. Moreover, work in humans suggests that although 'liking' and 'wanting' both represent a positive reward, they utilize distinct neural processing (*Wise, 2002*; *Schultz, 2006*). Our observations with D- and L-arabinose now provide an opportunity to explore the neural basis of 'liking' and 'wanting', and how these reward percepts strengthen memory in the accessible nervous system of *Drosophila*.

## Attributes of sugars important for long-term memory

The caloric value of a sugar has been found to be an important determinant of long-term appetitive memory (*Burke and Waddell, 2011*; *Fujita and Tanimura, 2011*; *Musso et al., 2015*), implying that flies quickly metabolize the sugar and that caloric evaluation somehow provides cues necessary to elicit long-term memory. We find that sugar with no caloric value can also produce long-term appetitive memories. One obvious possibility is that memories of sweet nutritious sugars are distinct from memories of sweet non-nutritious sugars. However, this seems so far not to be the case: a subset of higher order dopaminergic neurons (R58E02GAL4) necessary for long-term memory of nutritious sucrose (*Liu et al., 2012*) is also required for non-nutritious L-arabinose (*Figure 6—figure supplement 1B*). Similarly, addition of sorbitol, a tasteless but nutritious sugar, enhances the memory of non-nutritious sugars like xylose and D-arabinose, but does not enhance the memory of nutritious sugars (*Burke and Waddell, 2011*). Adding sorbitol to L-arabinose had no additive effect on long-term memory (*Figure 6—figure supplement 1C*). It therefore appears that L-arabinose memory uses at least some of the same downstream neural circuitry as memory of nutritious sugars.

Whether memory of L-arabinose, a non-nutritious sugar, is an exception or represents a more general phenomenon is unclear since we have tested only a limited number of sugars in a particular behavioral paradigm. However, in addition to L-arabinose, L-fucose can also produce memory (*Figure 1E*); both are components of the pectin in many fruits' cell walls (*Dick and Labavitch, 1989*; *Ahmed and Labavitch, 1980*). It is therefore possible that these sugars may signal some specific attributes of ripening fruit—ripening is accompanied by breakdown of the fruit's cell walls—although neither of these sugars are present in fruits near the concentrations (1 M) used in memory assays. Nonetheless, these observations suggest that flies can quickly assess salient features of sugars—a sort of leading indicator of nutritional value—without the sugar's metabolic breakdown. This

approach to memory formation may allow flies to quickly recognize and remember potential foods using specific cues, a time advantage that could be vital in natural contexts.

Do insects distinguish structurally similar sugars? The taste modality of insects, particularly *Drosophila*, is reported to have limited discriminatory power and be primarily based on the intensity of the stimuli as opposed to the chemical nature of the sugar (*Masek and Scott, 2010*). Indeed we find that, apart from flies' differential preference for various sugars at equal concentrations, for immediate and short-term behavior this is largely true. However, we did not observe any obvious correlation between immediate behavior and long-term memories: flies immediate preference is L-fucose > D-arabinose > L-arabinose > L-sorbose (*Figure 1D*); for short-term memory, L-sorbose = D-arabinose ≥ L-arabinose = L-fucose (*Figure 1E*); but in order of long-term memory score, L-arabinose ≥ L-fucose ≥ D-arabinose = L-sorbose (*Figure 1F*). These results indicate that while short-term responses are guided by palatability, long-term behavioral reponses are guided by additional attributes of the sugars. It is not yet clear why D-arabinose is a less effective stimulus. Since D- and L-arabinose are both sweet, they may generate positive sensations in a different manner, or perhaps D-arabinose carries a negative value that over time reduces the positive association formed initially (or dampens the behavioral output).

## Role of gustatory receptors in long-term appetitive memory

The gustatory receptors Gr5a, Gr43a, Gr61a, and Gr64a-f have been implicated in sugar detection (*Fujii et al., 2015*; *Freeman and Dahanukar, 2015*; *Dahanukar et al., 2007*; *Scott et al., 2001*; *Dunipace et al., 2001*; *Montell, 2009*; *Jiao et al., 2007*, *2008*; *Joseph and Carlson, 2015*). Although exactly which Gr receptors are responsible for detecting which sugar remains somewhat controversial, two features of sweet-sensing gustatory receptors are generally agreed upon: first, different gustatory neurons express a number of Gr receptors in unique combinations; second, more than one receptor is typically involved in detecting a sugar (*Fujii et al., 2015*). However, the physiological consequences of this combinatorial expression of semi-redundant gustatory receptors remain uncertain. This study raises the possibility that gustatory neurons in different locations, expressing unique combinations of receptors, are responsible for discriminating chemically similar sugars and eliciting different behavioral responses. Consistent with this idea, previous studies suggested that Gr43a neurons in the central brain monitor hemolymph fructose levels and modulate feeding behavior (*Miyamoto et al., 2012*), while we find that these neurons are dispensable for L-arabinose memory, and that peripheral Gr43a-neurons are likely sufficient to signal the presence of a rewarding sugar and generate associative memories. These differences likely arise from the locations of these neurons, differentially expressed receptors, the presence or absence of various co-receptors, and the second-order neurons to which these neurons project. Exactly which or how many Gr43a-, Gr61a-, and Gr5a-expressing neurons in the periphery are sufficient for L-arabinose memory is currently unclear.

We also find that activation of Gr43a-expressing neurons by ReaChR but not dTrpA1 is able to generate appetitive memory, while artificially activating a subset of dopaminergic neurons (R58E02GAL4) by heat (dTrpA1) or light (ReaChR) both led to long-term memory (*Figure 6—figure supplement 1A*). How a difference in activity at the sensory level is conveyed to higher-order neurons, and how that difference is interpreted by the higher-order neurons, remains unclear. More concretely, why is dTrpA1 activation of a subset of dopamine neurons sufficient to generate memory, but dTrpA1 activation of Gr43a-expressing neurons is not? One possibility is that the activity requirements of neuromodulatory systems are less stringent than those for sensory coding, and that temporal selectivity occurs before the signal reaches these dopamine neurons. Alternatively, recent studies have indicated that dopaminergic neurons are functionally diverse, and that distinct population of dopaminergic neurons are involved in appetitive associative memory (*Cohn et al., 2015*; *Huetteroth et al., 2015*; *Krashes et al., 2009*; *Berry et al., 2012*; *Aso et al., 2014*; *Yamagata et al., 2015*; *Berry et al., 2015*; *Musso et al., 2015*; *Schwaerzel et al., 2003*). These reports raise the possibility that differing sensory inputs could activate different subsets of dopaminergic neurons.

How can structurally similar sugars generate differential activation? It is likely that although these sugars bind to some of the same receptors, the relative affinity of the receptors vary. In this regard, the fly sweet taste system may be similar to that of the mammalian system, where a single heteromeric receptor (T1R2 and T1R3) is responsible for detecting a large number of sweet substances,

with multiple discrete ligand-binding sites in each receptor responsible for generating diverse responses (*Yarmolinsky et al., 2009*). We suspect that the differential engagement of multiple gustatory receptors leads similar chemicals to generate differential activation of the same neurons, and that differential activation and different ensembles of activated neurons allows higher-order neurons to decode the relevant features of sugars. We speculate that, at least in *Drosophila*, evaluation of a sugar's long-term salience may be encoded in the activation pattern of subsets of gustatory neurons, which allows rapid evaluation and remembering of nutritious food in complex environments.

## Materials and methods

### Fly stocks

Flies were generously shared by Dr. John Carlson (Gr43aGAL4-9/CyO; Gr61aGAL4-9/CyO; Gr5a-GAL4, Gr64GAL4), Dr. Hubert Amrein (UAS-Gr43a; Gr43a$^{GAL4}$, with first coding exon replaced with GAL4, serving as ΔGr43a and used in crosses for behavioral training), Dr. Anupama Dahanukar (Gr61a-null mutant, Gr64a-null mutants and Gr5a-null mutants), Dr. Toshihiro Kitamoto (UAS-Shibir-e$^{ts}$), and Dr. Paul Garrity (UAS-dTrpA1). The wild-type Canton-S flies were generously provided by Dr. Scott Waddell and Dr. Troy Zars. Other fly stocks were obtained from Bloomington Fly Stock Center (UAS-Kir2.1 RRID:BDSC_6595; UAS-GCaMP3 RRID:BDSC_32235; UAS-GCaMP6m RRID:BDSC_42748; UAS-ReaChR RRID:BDSC_53749; Gr64eGAL4 RRID:BDSC_57667; Gr64fGAL4 RRID:BDSC_57669; Gr64dGAL4 RRID:BDSC_57665; ΔGr64d/e RRID:BDSC_23628; ΔGr64f RRID:BDSC_27883).

### Sugars

Sugars were obtained from the following sources: D-arabinose, Sigma, cat#A3131-25G, lot# SLBB3223V,100M1365V and Fisher Bioreagents, cat# BP250425, lot# 114986; L-arabinose, Sigma, cat# A3256-100G, lot# BCBB3602V,098K0164 and USB Corporation, cat# 11406, lot# 4131874; L-sorbose, Sigma, cat# 85541, lot# BCBD8834V; L-fucose, Sigma, cat# F2252, lot# SLBB1522V; L-rhamnose monohydrate, Sigma, cat# R3875, lot# BCBD8824V; D-sorbitol, Sigma, cat# S1876, lot# 017 K0092; sucralose, Sigma, cat# 69293, lot# BCBF8524V; and saccharin sodium salt hydrate, Sigma, cat# S1002, lot# BCBF4560V; arabinogalactan, Food Science of Vermont, item# 026664342010.

### Two-choice feeding assay using dye

The two-choice tests were performed essentially as previously described (*Weiss et al., 2011*): 1- to 2-day-old male flies were collected in groups of 50, allowed to recover for 3 days, and food-deprived for approximately 22 hr in plastic tubes (VWR) containing kimwipes wetted with 3 ml of water. 1% agarose (Sigma) was mixed into 1 M sugar solution along with red or green food dye (1%, McCormick), and 15 μl drops were pipetted into 60-well minitrays (Thermo Scientific). A hole large enough to fit a funnel was melted into the lid, and the 50 flies were allowed to feed for 5 min in complete darkness, with tape covering the lid hole. At the end of 5 min, the color in their abdomens was assessed under a dissecting microscope, and flies were counted as eating a sugar if any dye was visible in their abdomens or thorax. Flies eating a mix of the two were scored half for L-arabinose, half for D-arabinose. Preference and detection indices were calculated as (number of flies eating sugar)/ (total number of flies). To rule out the color bias in the cases of choice between two sugars, half the experiments had the colors reversed. The feeding assay was carried out for 5 min, instead of a period of hours, because in the context of our particular behavioral paradigm the choices made by flies over a longer time period are not relevant.

### Radioactive feeding assay

Two-choice radioactivity experiments were performed as described above, with the addition of 1 μL of 1:5 diluted cytidine 5'-triphosphate [α−32P] (3000 Ci/mmol 10mCi/ml, 1MCi; PerkinElmer) into 1.5 ml 1 M sugar solutions without dye; again sugars were pipetted onto the 60-well microtiter plate. After the 5-min feeding, flies were immediately placed on dry ice blocks, and five flies chosen at random were placed in each scintillation vial (Denville Scientific), homogenized, covered with 5 ml LSC-

cocktail (ScintiSafe, Fisher Scientific), and counted by scintillation counter (LS6500; Beckman Coulter).

## Video monitoring of feeding assay

Video monitoring of feeding flies was performed using webcams (C160; Logitech). Four colorless drops of 1% agarose and 1 M sugar solution were placed on an empty 35 mM Petri dish (Falcon), one in each quadrant; two were L-arabinose and two were D-arabinose. Video was recorded for 30 min; trials in which the flies never found the sugars were discarded from analysis. Once the fly encountered a sugar solution, the behavior for next 5 min were quantified. We also examined the preference for other sugars, including sweet versus non-sweet sugars, to ensure that the experimental conditions did not influence the flies' choices.

## Antibiotic feeding

Antibiotic experiments were carried out by placing approximately fifty 1- to 3-day-old flies into plastic tubes with kimwipes and 3 ml of either 1 M sucrose or 1 M sucrose with 100 µg/ml kanamycin, 500 µg/ml ampicillin, and 50 µg/ml tetracycline, for 24 hr. The antibiotic concentrations were chosen based on previously published work (*Ridley et al., 2013*; *Brummel et al., 2004*; *Sultan and Baker, 2001*). Flies were subsequently transferred to tubes with either 3 ml water or 3 ml water with 100 µg/ml kanamycin, 500 µg/ml ampicillin, and 50 µg/ml tetracycline, for another 22 hr. They were then trained with 1M L-arabinose as described below.

## Survival assay

Survival curves were generated by placing fifty 3–5 day old flies in plastic tubes with kimwipes soaked in 2.5 ml of 1 M sugar solution. For each sugar solution tested, ten individual tubes were tracked, thus $n$ = 10 for each solution. The number of dead flies was counted at 12, 24, 36, 60, and 72 hr.

## Appetitive-olfactory conditioning

Olfactory training was carried out largely as previously described (*Krashes and Waddell, 2011*): 1- to 3-day-old flies were made hungry by placing groups of 50–70 flies in plastic tubes with kimwipes and tap water (time of starvation was determined by mortality rate: approximately 20–24 hr for homozygous lines; 24–30 hr for heterozygous crosses). Forty-seven microliters of 4-methylcyclohexanol (MCH; Sigma) and 42 µl of 3-octanol (OCT; Alfa Aesar) were separately diluted into two bubble humidifiers (B and F Medical) each containing 50 ml of mineral oil (Fisher Scientific); bubble humidifiers were connected in parallel by ¼-inch clear PVC tubing (VWR). 8 cm x 10 cm rectangles of filter paper (410, VWR) were soaked in water or 1 M sugar solution, and allowed to dry until the paper was damp, then rolled to fit tightly into the training tubes. Groups of 50–70 flies were moved into the t-maze, then into the water tube for 2 min while MCH odor passed through, moved back to the holding chamber in the t-maze for 30 s, then moved to the sugar tube for 2 min while OCT odor was flowing through. The next group of flies was trained reciprocally, where OCT was paired with water and MCH with sugar. Unless otherwise specified, after training flies were fed for 4 hr and restarved until testing 24 hr after training. Flies were tested by being given a choice between OCT and MCH in tubes with no filter paper; test duration was 2 min. Short-term memory was assayed 2 min after training. Memory index = [(number of flies in reward odor – number of flies in unrewarded odor)/ (total number of flies)]. A memory index was calculated for each of the two reciprocal trials and then averaged; this average constituted an $n$ of 1. Sucrose was frequently used as a daily standard, thus the large numbers of sucrose trials. For experiments with two or more controls, the experimental line was trained in parallel with one of the controls, and then again trained in parallel with the other control—thus the large $n$ for both ChaGAL80 and ReAChR experiments.

## Split-GAL4 construct

The split-GAL4 vectors (*Pfeiffer et al., 2010*) were made using the pHD-ScarlessDsRed vector (*Gratz et al., 2014*), DGRC #1364. To construct the Gr43a-VP16 vector, the 5' homology arm was inserted into the AarI restriction site by Gibson assembly (GACTGAACCGTGTAGGGA . . . TCCCGCGTTCTGAATTACT), immediately followed by the VP16 sequence (ATGGATAAAGCGGAA

TTAATTCC . . . CTGGGCGGCGGCAAGTAA) (addgene #26268). The 3' homology arm was inserted into the SapI site (AGTAGTGACACTCGGA . . . GAAGACCATATACGTC). CRISPR oligos were designed using http://tools.flycrispr.molbio.wisc.edu/targetFinder/ (target sequences: AGAAC TGGGACCTTACAAGT and TACCTACCGCACGGGAATTT). To construct the Gr5a-DBD vector, the 5' homology arm: ACTTCGTTTGGCGTTTC . . . TAGAGCTTGTACACA, followed immediately by the GAL4 DNA-binding domain sequence (ATGCTGGAGATCCGC . . . ACAGTTGACTGTATCGTAA). The 3' homology arm for the Gr5a-DBD vector: ATGATGCTTTTCTTCGC . . . TCAACGGCCGTGC TCCTCT. CRISPR target sequences for the Gr5a locus were TGATTCCACACACGGGCATT and CGCACATCCAGCACACTGT. CRISPR gRNAs were ligated into the pU6-BbsI-chiRNA (addgene #45946) and pU6.2-BbsI-chiRNA vectors(*Gratz et al., 2013*). DNA was mixed at a ratio of pHD-dsRed 500 ng/ul to U6-gRNA 100 ng/ul, and injected by BestGene, Inc. at a final concentration of 250 ng/ul.

## Optogenetic and TrpA1 stimulation

Optogenetic activation was performed using the same hardware as previously published (*Inagaki et al., 2014*), except that two rows of six LEDs each were aligned parallel to the tube, 2 cm away, at 90° angles to each other. To minimize behavioral artifacts caused by strong visual stimulation, the red (627 nm, 161 lm @ 700 mA) Rebel LEDs were chosen, and the stimulation protocol (pulse width, intervals, and duration) was controlled by Arduino board and Arduino computer language. For dTrpA1 experiments, the relevant training tube was preheated to 31°C, and during training was wrapped in a ReptiTherm Under Tank Heater (RH-4; Zoo Med Laboratories); the temperature was held constant (at 31°C) by an electric temperature control with probe placed in between the wrapped layers (A419; Johnson Controls). The heater temperature required to maintain an internal tube temperature of 30°C was determined empirically.

## Statistical analysis and number of trials (*n*)

All statistical analyses were performed using Graphpad Prism 5. All the data met the assumption of homogeneity of variance, therefore unpaired two-tailed t-test or one-way analysis of variance (ANOVA) was performed with Tukey post-hoc test between pairs of samples. ANOVA tests for significance were performed a probability value of 0.05 and more stringent values are listed in each figure where applicable. For all experiments, each *n* is considered a biological replicate; separate trials used independent samples of genetically identical flies. For two-choice experiments, a single *n* constitutes a population measure generated from 50 male flies. The preference index indicates the proportion of flies eating the sugar, which was determined by scoring visible color in the abdomen or thorax. For video monitoring, each *n* constitutes a single fly. For survival curves, each *n* is a population measure generated by 50 flies placed in a tube with 1M sugar. Percent survival indicates the percentage of flies alive at each timepoint. For olfactory training experiments with sugar, heat, and light: one trial consists of giving a group of approximately 50–70 flies water and 3-octanol for 2 min, waiting 30 s, then giving sugar and 4-methylcyclohexanol for 2 min. Another group is trained with water and 4-methylcyclohexanol, then sugar and 3-octanol. Memory indices are calculated for each of these two trials and averaged. This average constitutes a single *n*, which is approximately 100–140 flies. Based on the previous and ongoing experimental effect sizes, 8–10 of these double trials were generally judged to be adequate for memory experiments, unless effect sizes were strikingly large or variable. The more dramatic effect sizes and smaller variability of preference assays allowed a smaller number of trials, generally 4. In all long-term memory experiments, experimental manipulations for which a negative result was plausible or expected were always trained alongside a positive control. This is the reason for conspicuously large numbers of trials with sucrose and L-arabinose compared to other sugars or manipulations. Similarly, for experimental groups needing to be compared to two or more controls, the experimental group was first trained alongside one of the control groups, and then again trained alongside the other control group (s). This is the reason for large numbers of trials in, for example, the ChaGAL80 and ReaChR experiments.

## Immunostaining

Tissues were dissected in PBS (Sigma), and fixed in 4% paraformaldehyde (Electron Microscopy Sciences) in PBS-Triton. 3% (PBST) (Sigma) for 1–2 hr. They were washed in PBS-Triton: 3% five times

for 15 min each time, and blocked in PBST with 10% normal goat serum (Vector Laboratories) for 2 hr. Rabbit anti-GFP IgG (MBL International Corporation) was diluted 1:1000 in the blocking solution and centrifuged at 14,000 r/min for 10 min at 4°C. Tissues were incubated with primary antibody overnight at 4°C, then washed again with PBST for 15 min, five times. Anti-rabbit IgG Alexa Fluor 488 (Life Technologies, now ThermoFisher Scientific, Waltham, MA) was diluted 1:1000 in blocking solution, and incubated with the tissues overnight. Tissues were again washed five times, and mounted in Vectashield (Vector Laboratories) on slides with doubled clear reinforcement labels (Avery); No. 1 ½ coverslips were used (VWR). Images were acquired on a Zeiss Pascal confocal microscope with a Plan Apochromat 20 × 0.8 NA objective. GFP fluorescence was excited at 488 nm and emission was collected through a 505–530 nm bandpass filter.

## Calcium imaging

Tissues from Gr43a$^{GAL4}$ x UAS-GCaMP3 or UAS-GCaMP6med flies were prepared largely as described previously (*Miyamoto et al., 2012*). Two- to 7-day-old flies were used. All tissues were dissected in Ringers solution (5 mM HEPES, 130 mM NaCl, 5 mM KCl, 2 mM CaCl$_2$, 2 mM MgCl$_2$); legs were removed from the fly, placed on a 50 mm glass-bottomed dish No 1.5 (Mattek), and immobilized with a 1.5 µl drop of 2-hydroxyethylagarose (Sigma). After the agarose firmed, 20 µl of Ringers was added to cover the leg. In D- vs L-arabinose comparisons, both front legs of the fly were used as matched controls. Brains adhered to the dish without need for agarose when placed into a 30 µL bubble of Ringers. Proboscis imaging was performed with the proboscis upside down on the plate, so that the dorsal proboscis was contacting the dish; the proboscis was immobilized with 1.5 µl of agarose and again covered with 30 µl of Ringers solution. Only one sugar was tested per tissue sample. Images were collected at least 40 s before sugar was added; sugar was added at 2x concentration, in the same volume as the Ringers covering the tissues. Because the training paradigm uses high concentrations (up to 1 M) of sugar, we used 500 mM sugar concentrations for the leg and proboscis imaging. However, for the brain, 500 mM appeared to cause osmolarity-induced shrinking, so brain imaging used 100 mM sugars. Leg imaging was performed at approximately one stack per 5–7 s; proboscis imaging at approximately one stack per 13 s; brain imaging at approximately one per 14 s. Only tissues that showed a response were used in analysis, although tissues that didn't respond were checked for viability by adding fructose as a positive control. Images were acquired on a Zeiss Pascal confocal microscope with a Plan Apochromat 20 × 0.8 NA objective. GFP fluorescence was excited at 488 nm and emission was collected through a 505–530 nm bandpass filter. For calcium imaging of the leg with ΔGr43a-GAL4 x UAS-GCaMP3 and ΔGr43a-GAL4; UAS-GCaMP6med flies, images were acquired on an Ultraview Vox (PerkinElmer) with a Plan Apochromat 20 × 0.8 NA objective at approximately one stack per 10 s; GFP fluorescence was excited at 488 nm and collected through a 525–550 nm bandpass filter. Analysis was performed in ImageJ (NIH) using in-house plugins: z-stack images were sum-projected and camera background was subtracted by selecting a region of interest away from the tissue. Where needed, the StackReg registration plugin was used to minimize movement artifacts (*Thévenaz al., 1998*). Measurements were always taken by encircling cell bodies. In tissues with more than one neuron visible, the response of each neuron was analyzed separately and then averaged to generate an average response for that single tissue; this average constituted a single *n* and was used with others to generate average response curves and peak ΔF/F$_o$. Peak ΔF/F$_o$ measurements were made by taking the first peak value, and dividing by the average of five timepoints immediately preceding the rise. To generate normalized fluorescence curves, individual tissue averages were aligned by the first timepoint of the rise. Curves for leg and proboscis were linearly resampled at 3 s; brain at 5 s. Curves were then min/max normalized, and average trajectories were calculated. Error bars were calculated as standard error in the mean. Average curves were plotted in GraphPad Prism 5.

## CAFÉ assay

One-day-old Canton S adult flies (males and females) were transferred to fresh standard food medium for 1 day and then starved (with free access to water) for 18–22 hr. These flies were then transferred by groups of 20 into plastic boxes (*Sellier et al., 2011*). Each box had a row of five capillary tubes (5 µl minicaps, Hirschmann LaborGeräte, Germany), filled with a dilution of sugar mixed with a red dye (erythrosine 0.374 mg/ml; Sigma France). The concentrations of sugar (L- and

D-arabinose, Sigma, France) were: 1 M, 100 mM, 10 mM, 1 mM and 0 mM. Each box was monitored with a webcam (HD Pro C920 or QuickCam Pro 9000, Logitech). The boxes and cameras were housed in a climatic chamber maintaining a temperature of 25°C and 80% H.R. (DR-36 VL, CLF Plant Climatics GmbH, Germany). For each box, images were acquired at a rate of one image/ min for 2 hr using the software VisionGS, Germany. The stack of images was then transferred to ImageJ (*Abramoff et al., 2004*) and the liquid level of each capillary was analyzed using a Java plugin, and subsequently transferred to Excel. Results are expressed as the mean of the change of the liquid level in each capillary (D-arabinose: n = 12; L-arabinose: n = 10 boxes). Error bars are computed as the standard error to the mean (s.e.m.).

## Electrophysiological recordings

Tip-recording was performed as previously described (*French et al., 2015*). Briefly, adult flies (3- to 4-day old) were anesthetized on ice and immobilized on a putty platform (UHU stick), using thin stripes of tape. They were then disposed under a stereomicroscope (MZ12, Leica) and specific sensilla from the proboscis or from the legs were stimulated and recorded, using a TasteProbe amplifier (DT-02, Syntech, Germany; *Marion-Poll and Pers, 1996*) connected to a general purpose amplifier (CyberAmp 320, Data Translation, USA) which further amplified (x100) and filtered the signal (10 Hz-2800 Hz). The stimulus electrode contained tricholine citrate (TCC 30 mM), in order to allow an electric contact to be established with the sensillum and to inhibit firing activity arising from water-sensitive cells (*Wieczorek and Wolff, 1989*). A reference electrode was connected to the abdomen of the fly, using a drop of electrocardiogram gel. Each stimulation lasted 2 s and was digitized at 10 kHz, 16 bits during 2 s (DT9818, Data Translation, USA). The data acquisition, spike detection and sorting was performed under a program, dbWave. The results were subsequently transferred to Excel, and expressed as the mean (n = 8–15 measures). Error bars were computed as the s.e.m.

## Acknowledgements

KS and JM conceived of the project, designed the experiments, analyzed the data, and prepared the manuscript. JM, HJ, AVA., and CPS performed the experiments. JU, BS, and JL assisted with imaging and image analysis. ZY built the optogenetic apparatus. KS and JM wrote the paper and made the figures. MAA performed the electrophysiological observations, Palak Rawat performed CAFE experiments, and FMP analyzed the CAFE and electrophysiology data. We are especially grateful to Dr. John Carlson of Yale University for reagents and suggestions in the early stages of the project. We thank Wanda Colon Cesario, Adrienne van Antwerp, and Abiel Trevino for assistance with sugar preference tests and calcium imaging. We thank Tony Torello for help in constructing the optogenetic apparatus. We thank Liying Li, Amitabha Majumdar, Gunther Hollopeter, Jay Unruh, and Brian Slaughter for important discussions. This work was supported by the Stowers Institute for Medical Research.

## Additional information

### Funding

| Funder | Grant reference number | Author |
| --- | --- | --- |
| Stowers Institute for Medical Research | SIMR funding | Kausik Si |

The funders had no role in study design, data collection and interpretation, or the decision to submit the work for publication.

### Author contributions

JPM, KS, Conception and design, Analysis and interpretation of data, Drafting or revising the article; HJ, Conception and design, Acquisition of data, Analysis and interpretation of data; MAA, CPS, JL, ZY, Acquisition of data, Analysis and interpretation of data, Drafting or revising the article; FM-P, Conception and design, Acquisition of data, Analysis and interpretation of data, Drafting or revising the article

Author ORCIDs
Kausik Si, http://orcid.org/0000-0002-9613-6273

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
