## [Decision Letter]

[Editors’ note: a previous version of this study was rejected after peer review, but the authors submitted for reconsideration. The first decision letter after peer review is shown below.]

Thank you for submitting your article "Immediate perception of a reward is distinct from the reward's salience" for consideration by *eLife*. Your article has been reviewed by three peer reviewers, one of whom, Mani Ramaswami, is a member of our Board of Reviewing Editors, and the evaluation has been overseen by a Senior Editor.

The reviewers have discussed the reviews with one another and the Reviewing Editor has drafted this decision to help you prepare a revised submission. However, we note that the required changes are extensive and may take longer than the two months we normally allow for return of a revised submission. Therefore, we ask that you respond to this request with a letter detailing the experiments you are prepared to complete and an estimate of the time you expect this will take. We will send a recommendation when your responses are evaluated by the Board.

Summary

McGinnis et al. investigate an interesting question in this study, of why two structurally related sugars D-arabinose and L-arabinose engender different outcomes in an appetitive learning paradigm. They explore differences in hedonic value (sweetness) and find that L-arabinose, which is less preferred in residence or feeding assays, can form long-term associative memory, whereas D-arabinose cannot. Therefore, the study makes the unexpected observation that the strength of the immediate gustatory response of a fly to a sweet compound does not necessarily predict its strength as a reinforcer for associative memory. Thus, 1M D-Arabinose which is perceived as much sweeter than 1M L-Arabinose by the fly in a simple behavioural test, is remarkably much weaker than the L-isomer as an unconditional reinforcing stimulus in an appetitive conditioning procedure that induces LTM.

This is striking and suggests different levels at which the quality of a tastant is encoded in the brain. One level is its immediate attractiveness. And a second independent level its reward value as used in the encoding of associative memory. Because neither sugar has any nutritional value, this analysis differs from related recent work of Tallez et al. Nature Neuroscience 2016 (which really should be cited) and Heutteroth et al. Curr. Biology 2015 (which is cited here), both of which have characterized different central mechanisms of encoding sweetness and nutrient value of sugars in mouse and *Drosophila* respectively. In contrast McGinnis et al. suggest that "sweetness" and long-term "reinforcement value," of sugars are evaluated distinctly of each other as well as of nutritive value.

The authors then perform a range of experiments on the basis of which they argue that the two qualities are discriminated to some degree at the level of sweet sensory neurons. This would be a novel conclusion, particularly given the conclusions of Masek and Scott 2010, who postulate that sweet sensory neurons can only encode relative sweetness (or sugar concentrations) and not qualitative differences between sugar compounds. The authors use a panel of Gr-GAL4 drivers to identify neurons that function in L-arabinose memory, use calcium imaging to characterize D-ara and L-ara responses in a pair of Gr43a tarsal neurons, and use optogenetics to show that pulsed activation, but not continuous activation, of these neurons can form long-term memory (LTM) in the absence of sugar reward in hungry flies. Their genetic and behavioural data suggest that the Gr61a receptor expressing neurons contribute to sweetness discrimination and that a combinatorial effect of Gr61 and Gr43a expressing neurons is involved in determining reinforcement strength. Although these initial findings are very interesting and the study encompasses a large number of experiments, in its present form they fail to deliver conclusive results that support the model that variation in D-ara and L-ara evoked activity in Gr43a neurons is the mechanism underlying the differential effects of these sugars in generating LTM.

Major issues and comments that must be addressed:

Of several comments and observations made below, points 1-6a must be addressed experimentally. The revised manuscript must include experiments should at least strongly and conclusively show that this discrimination occurs at the level of sensory neurons through a mechanism that is different from the perceived level of sweetness. Measurements of D- and L-ara responses across a variety of concentrations, with PER assays and perhaps electrophysiology tip recordings will be necessary. The discussion in this context should include Masek and Scott, (PNAS) 2010, whose hypotheses and findings we feel should be cited prominently and tested experimentally. The authors should take up this challenge before submitting a revised paper.

Regarding points 6b-8, we appreciate that it will be even more challenging to identify the exact mechanism by which sweetness and reinforcement strength are encoded and differentiated. In our view, given the difficulty with showing that the ChR stimulus regimens satisfactorily approximate respective and relevant sugar stimulation, it may be best to simply drop this section from the manuscript.

If the paper is revised as we suggest, then various statements in the Introduction and Discussion will also need to be revised.

1) Figure 1 makes the amazing observation that while flies prefer to eat D-arabinose, they form associations better with L-arabinose. There are potential simple physiological explanations for this that should be addressed.

a) That D-arabinose being eaten in larger quantities makes flies feel ill and therefore has the effect of being a part negative reinforcer for LTM.

b) That the expectation of nutritive value is higher for D-arabinose and a "dopamine prediction error" type mechanism comes into play, which results in D-arabinose having a long-term negative valence even though it is highly positive initially.

Both of the above possibilities could be partly addressed if the LTM experiments are performed at concentrations at which the two sugars are equally attractive (consumed at similar levels, or showed similar levels of PER). From Figure 1, this would likely correspond to a 1M conc of L-arabinose and maybe 200-250mM conc of D-arabinose. A possibly instructive finding would be that as concentrations of D-arabinose are reduced, its ability to act as an LTM reinforcer would increase. Tests at varied concentrations of D-arabinose that bracket this concentration would be useful.

2) The effect of different receptor mutants on LTM to D-arabinose should also be tested. A prediction is that reduced immediate attraction to D-arabinose may increase D-arabinose LTM, but reduce L-arabinose LTM (the latter is shown).

3) Although the authors conclude that a subset of taste neurons is critical for associative memory with a specific sugar, a specific subset of neurons has not been definitively identified. The statement that "it appears that the distal tarsi Gr43a-Gr61a- expressing neurons that also express Gr5a are critical for L-arabinose memory" is weak. Given that GAL4 and LexA drivers are available for both Gr5a and Gr43a, it is possible to manipulate only the overlapping neurons of interest to determine if they are indeed critical for L-arabinose memory. This is particularly important because of variability in GAL4 driver expression, and results that conflict with expectations from previous mapping analyses: for example, previous reports map Gr43a-GAL4 neurons as a subset of Gr61a-GAL4 neurons. Yet, pulsed activation with Gr43a-GAL4 is sufficient for long-term memory but that of Gr61a-GAL4 is not. It would be useful to explain this discrepancy. One of those studies also maps Gr5a-GAL4 and Gr43a-GAL4 to mutually exclusive subsets of tarsal neurons.

4) The observation that hedonic value of a sugar can be unlinked from its potential to generate associative memory is very interesting. However, inferring detection (sweetness) solely from feeding choice experiments can be problematic – for instance, feedback during and post-ingestion is still possible, even in 5-minute assays because of differences in pharyngeal input (as reported here) and Gr expression in the gut (as reported previously). Proboscis extension assays would serve as a more immediate read-out of degree of instant "liking" and should be performed by stimulating either labellar or tarsal hairs.

5) In the same vein, binary choice assays are not the optimal tool to examine discrimination because loss of sensitivity for both sugars could confound the results (supported by the greater number of flies that do not participate). It would be more useful to determine threshold concentrations for D-ara and L-ara behavioral responses against water. This has been done to some extent for L-arabinose in Figure 2—figure supplement 1, but the analysis with a single concentration may not uncover functional redundancies in L-ara detection between the different Gr-GAL4 neurons that are being tested. This is supported by the observation that Gr43a neurons respond to L-ara (Figure 3) but are not required for behavioral response to L-ara (Figure 2—figure supplement 1). To really make the following statements: "Taken together these results suggest that preference for D-arabinose relies on Gr43a and Gr61a-expressing neurons, while L-arabinose memory relies on Gr43a-, Gr61a-, and Gr5a expressing neurons. However, D-arabinose preference can be maintained in the absence of any single receptor (Figure 2), whereas L-arabinose memory specifically requires Gr43a and Gr61a receptors (Figure 2)," it is necessary to test various concentrations of D-ara versus water to conclude that D-ara preference is maintained in the absence of any receptor. We suspect that this will change with the mutants tested as they will impact how "sweet" D-ara is, and thus, how much flies ingest of it. It is not quite enough to test D-ara preference relative to L-ara. That could simply reflect relative sweetness.

6a) The observation that two structurally related sugars can evoke different temporal patterns of GCaMP activation in the same neuron is novel, but needs to be explored further. With the GCaMP analysis provided, the differences in activation patterns evoked by D-ara and L-ara in Gr43a neurons are not entirely convincing and have not been investigated to the extent that is necessary. In particular, the authors should examine these patterns across concentrations, and quantify the temporal differences in activation patterns rather than presenting only peak values (which are not different). Ideally, tip-recordings would provide higher resolution and more convincing data to determine if L-ara and D-ara are eliciting distinct neural patterns that could allow flies to discriminate between the 2 molecules. In this scenario, dose responses of the two sugars should be tested with tip recordings of proboscis taste sensilla (or leg taste sensilla).

6b) Additionally, it would be useful to know how the response patterns of D-ara and L-ara compare with those of sucrose and fructose, which are canonical activators of these neurons. Also, what is the role of Gr43a in the temporal kinetics of the response (as opposed to the strength of the response)?

7) The conclusion that different patterns of activation within the same neurons, rather than activation of different sets of neurons, may be important for different behavioral outcomes is based on pulsed activation using optogenetics. A major gap that remains is the question of what pulsed activation mean in terms of calcium activity, and how that connects with possible differences in activation patterns evoked by D-ara and L-ara. Also, pulsed activation of both Gr5a-GAL4 and Gr43a-GAL4 can form LTM, but previous evidence suggests that they are not expressed in the same subsets of neurons.

8) In the same section, "To test the possibility that long-term memory requires activation of sensory neurons in a more temporally refined manner, we used the red-shifted channelrhodopsin variant ReaChR" – it is crucial to know whether pulsed stimulation gives rise to the same number of spikes than a (chopped) continuous pulse. From the study of Inagaki et al., it seems that despite the fact that ReaChR triggers more tonic responses, it is still quite phasic. A pulsed stimulus could be more efficient in terms of total number of action potentials generated. In other words, this experiment is not conclusive regarding the frequency of the stimulation, as it can be read also that patterned stimuli are more efficient simply because more spikes are generated. This assumption is wrong (same paragraph): "For example, certain pulsated patterns of light (20 Hz, 15 msec pulse-width for 2.5 ms, repeated every 20 seconds) generated long-term associative memories, while the same amount of light delivered in a uniform, tonic pattern was markedly less effective (15 msec pulse-width repeated every 327 msec continuously) (Figure 4)." – it could be less effective because less spikes are generated. The "amount of light" is not a trustworthy indication of how much spiking is generated in the neurons.

[Editors’ note: what now follows is the decision letter after the authors submitted for further consideration.]

Thank you for resubmitting your work entitled "Immediate perception of a reward is distinct from the reward's long-term salience" for further consideration at *eLife*. Your revised article has been favorably evaluated by Mani Ramaswami, Reviewing editor, and one other reviewer.

The manuscript has been improved but there are some remaining issues that need to be addressed before acceptance, as outlined below:

Reviewer #1:

The author's extensive revisions add considerably to the validity and interest of the conclusions made in this manuscript. This new submission far more convincingly demonstrates the key point of their manuscript, that the immediate perception of a reward is distinct from the reward's long-term salience. Most important, by testing a range of concentrations of L and D arabinose, the new data show that the relatively high reinforcement value for L-Arabinose compared to D-Arabinose for LTM (but not STM) cannot be explained simply based on the relative amounts consumed and differences in associated toxicity. As noted previously, this unexpected behavioural observation is striking and should be of interest both from a fly learning and memory perspective and to psychologists interested in learning theory.

Less clear is the mechanism by which LTM reinforcement value is encoded. The authors use behavioural and physiological (including tip recording) to identify subsets of tarsal sensory neurons that contribute to L-Arabinose response as well as f5v sensory neurons that show differential responses to D vs L arabinose and required for memory formation. This is interesting. However, since the neurons respond more strongly to D arabinose, than L, it still leaves open the question of which sensory neuron type communicates information on L-Ara quality and concentrations that determine its reinforcement value for LTM. Particularly given their already herculean efforts to identify the key L-Arabinose-responsive cells, a resolution of this issue seems outside the scope of this particular paper.

Essential revision

The authors should clearly acknowledge that the mechanism by which sensory neurons communicates information on L-Ara quality and concentration as relevant to LTM remains unanswered. This important ambiguity should be clearly stated in all relevant parts of the text and acknowledged by addition of a single line in the revised Abstract.

Recommended revisions (not essential, but to be considered seriously).

1) The following is a suggestion for a presentation that may be easier for the non-expert reader.

The current organisation of the almost overwhelming amount of data make it very difficult to read the manuscript and the reader is often disoriented by having to go back and forth between different figures. The reviewers suggest a reorganization of the figures in the following order.

Figure 1: Immediate preference for a sugar is not predicative of long-term memory formation – different sugars from fruits. Figure 2: Immediate preference for a sugar is not predicative of long-term memory formation – L- vs D-arabinose. Figure 3: The roles of Gr5a, Gr43a and Gr61a expressing neurons in immediate preference and memory formation. Figure 4: Peripheral Gr43a expressing neurons in the foreleg are critical for memory formation. Figure 5: Evoked-calcium activity and spike response of the f5v neuron in response to L- and D-arabinose. Figure 6: Activation of the Gr43a neurons mimics sugar stimulation in memory formation.

2) The interpretation of some of observations from the receptor mutants is difficult, because a few of are imprecise mutations. It may be best to only retain Gr43a and Gr61a mutant data, which make the authors' points. The essential data crucial for the paper is about which neurons mediate memory formation. For this reason, the GAL4-mediating silencing experiments are important and the Gr43a and Gr61a mutations augment the argument. The rest are potentially distracting. This point applies to the current Figure 2 and all supplemental information.

3) For most of the figures, ANOVA followed by the Tukey's test was used. Typically, statistical significance was denoted by using different letters such as a, b, and c.

4) Figure 1—figure supplement 2. For panel C, few people have the working memory to figure out the preference order of the different sugars. Using a matrix plot of three columns (D-arabinose, L-arabinose, and L-sorbose) and three rows (L-fucose, D-arabinose, and L-arabinose) for the preference comparison would provide a better illustration. Furthermore, the established order should be maintained in the panels D and F. A direct plot of memory index against immediate preference would be even better.

5) Figure 1—figure supplement 3. The panel order is not matched with that in the legend.

6) Figure 3—figure supplement 1. The inconsistence with the published paper (Fujii et al., 2015) needs to be mentioned. Figure 2 in that paper indicates that the f5v neuron expresses the Gr43a receptor but not Gr5a. However, here the authors found that the f5v neuro expresses both Grs.

7) Introduction. Confusing references. "A specific illustration of this phenomenon is seen with the two chemically similar sugars, D- and L-arabinose: flies greatly prefer D-arabinose to L, but form long-term memories of L-arabinose (a sugar released from pectin as fruit ripens) more consistently (Ahmed and Labavitch, 1980, Dick and Labavitch, 1989)". The references cited discuss L-arabinose being a sugar in ripened fruits and have nothing to do memory formation.

8) It would be helpful to have a graphical summary of the different Gr-GAL4 lines in the various neurons.

---

## [Author Response]

[Editors’ note: the author responses to the first round of peer review follow.]

*[…] 1) Figure 1 makes the amazing observation that while flies prefer to eat D-arabinose, they form associations better with L-arabinose. There are potential simple physiological explanations for this that should be addressed.*

*a) That D-arabinose being eaten in larger quantities makes flies feel ill and therefore has the effect of being a part negative reinforcer for LTM.*

*b) That the expectation of nutritive value is higher for D-arabinose and a "dopamine prediction error" type mechanism comes into play, which results in D-arabinose having a long-term negative valence even though it is highly positive initially.*

*Both of the above possibilities could be partly addressed if the LTM experiments are performed at concentrations at which the two sugars are equally attractive (consumed at similar levels, or showed similar levels of PER). From Figure 1, this would likely correspond to a 1M conc of L-arabinose and maybe 200-250mM conc of D-arabinose. A possibly instructive finding would be that as concentrations of D-arabinose are reduced, its ability to act as an LTM reinforcer would increase. Tests at varied concentrations of D-arabinose that bracket this concentration would be useful.*

As suggested we have measured the D-arabinose memory at lower concentration bracketing the 250mM concentration. Lowering the sugar concentration did not lead to higher memory score. (Figure 1—Figure supplement 4G; last paragraph in “*Drosophila melanogaster* prefers D-arabinose...” section.) Other work has also demonstrated that flies’ consumption of D-arabinose is not greater than more memorable sugars, such as D-glucose (Fujita and Tanimura, 2011).

*2) The effect of different receptor mutants on LTM to D-arabinose should also be tested. A prediction is that reduced immediate attraction to D-arabinose may increase D-arabinose LTM, but reduce L-arabinose LTM (the latter is shown).*

We have tested D-arabinose memory in Gr receptor mutant and following silencing of Gr- receptor expressing neurons. (Figure 2—figure supplement 2; last paragraph of “Gustatory neurons and receptors involved...”.)

*3) Although the authors conclude that a subset of taste neurons is critical for associative memory with a specific sugar, a specific subset of neurons has not been definitively identified. The statement that "it appears that the distal tarsi Gr43a-Gr61a- expressing neurons that also express Gr5a are critical for L-arabinose memory" is weak. Given that GAL4 and LexA drivers are available for both Gr5a and Gr43a, it is possible to manipulate only the overlapping neurons of interest to determine if they are indeed critical for L-arabinose memory. This is particularly important because of variability in GAL4 driver expression, and results that conflict with expectations from previous mapping analyses: for example, previous reports map Gr43a-GAL4 neurons as a subset of Gr61a-GAL4 neurons. Yet, pulsed activation with Gr43a-GAL4 is sufficient for long-term memory but that of Gr61a-GAL4 is not. It would be useful to explain this discrepancy. One of those studies also maps Gr5a-GAL4 and Gr43a-GAL4 to mutually exclusive subsets of tarsal neurons.*

For concerns about coexpression of Gr5a and Gr43a, Miyamoto et al. 2012 show that Gr43aGAL4 and Gr5aLexA are coexpressed in one tarsal neuron (and a few neurons in the labellum). Nevertheless, the reviewer is correct that these receptors label two largely distinct sets of neurons, and it is possible that either of these sets are sufficient, when activated, to generate appetitive memory. To address this issue we tried to use LexA/GAL4 flies to turn off GAL4 expression in the dual positive neurons (LexAop-GAL80). Since silencing of all Gr43a positive neurons impairs memory, we intended to test whether just de-silencing of the Gr43a-Gr5a neurons would be sufficient to form L-ara memory; all remaining Gr43a-expressing neurons would remain silenced. Unfortunately, with the stocks we have, flies bearing Gr43aGAL4::UAS-Kir; Gr5aLexA::LexAop-GAL80 transgenes were not very healthy and unsuitable for long-term memory assay. The LexA-GAL80 system would in any case not allow us to determine whether activation of just these dual positive neurons are sufficient for long-term memory.

Keeping this in mind we have generated a split-GAL4 fly expressing one half of GAL4 (VP16- activation domain) from the endogenous Gr43a locus and the other half (DNA-binding domain) from the Gr5a locus. When these flies were crossed to UAS-CD8EGFP flies a single neuron in the distal tarsi was labeled, consistent with the possibility that there is a single Gr43a-Gr5a positive neuron (Figure 3—figure supplement 1). However, in ~50% of the flies we observed additional neurons in the leg, and all flies had many labellar proboscis neurons labelled. The labelling of other neurons in addition to the distal tarsal neurons render these flies unsuitable for the memory assay, since a group of flies are trained in the appetitive-associative memory and variation among individual animals makes the population result difficult to interpret. (Figure 3—figure supplement 1 and please see appendix Figure 1)

*4) The observation that hedonic value of a sugar can be unlinked from its potential to generate associative memory is very interesting. However, inferring detection (sweetness) solely from feeding choice experiments can be problematic – for instance, feedback during and post-ingestion is still possible, even in 5-minute assays because of differences in pharyngeal input (as reported here) and Gr expression in the gut (as reported previously). Proboscis extension assays would serve as a more immediate read-out of degree of instant "liking" and should be performed by stimulating either labellar or tarsal hairs.*

As the reviewer has suggested we have performed proboscis extension assays for D-ara and L- ara at various sugar concentrations, and they are very similar. (Figure 1—figure supplement 4D)

*5) In the same vein, binary choice assays are not the optimal tool to examine discrimination because loss of sensitivity for both sugars could confound the results (supported by the greater number of flies that do not participate). It would be more useful to determine threshold concentrations for D-ara and L-ara behavioral responses against water. This has been done to some extent for L-arabinose in Figure 2—figure supplement 1, but the analysis with a single concentration may not uncover functional redundancies in L-ara detection between the different Gr-GAL4 neurons that are being tested. This is supported by the observation that Gr43a neurons respond to L-ara (Figure 3) but are not required for behavioral response to L-ara (Figure 2—figure supplement 1). To really make the following statements: "Taken together these results suggest that preference for D-arabinose relies on Gr43a and Gr61a-expressing neurons, while L-arabinose memory relies on Gr43a-, Gr61a-, and Gr5a expressing neurons. However, D-arabinose preference can be maintained in the absence of any single receptor (Figure 2), whereas L-arabinose memory specifically requires Gr43a and Gr61a receptors (Figure 2)," it is necessary to test various concentrations of D-ara versus water to conclude that D-ara preference is maintained in the absence of any receptor. We suspect that this will change with the mutants tested as they will impact how "sweet" D-ara is, and thus, how much flies ingest of it. It is not quite enough to test D-ara preference relative to L-ara. That could simply reflect relative sweetness.*

The primary reason we have used such a high concentration of D- and L-ara is because this is the standard concentration of sugar used in appetitive-associative memory training (by us and others); this also allowed us to compare our results with published studies. Nonetheless we agree concentration is a key factor and as suggested we have examined the ability of starved flies to detect D-arabinose and L-arabinose over water at various concentration of sugars. We found at 50mM, L-arabinose detection drops significantly but the flies still detect D-arabinose. However when the concentration drops to 10mM D-arabinose detection is also reduced. Unlike L-and D- arabinose the flies can detect 10mM sucrose at a similar level to that of 1M. Interestingly, however, at 10mM concentrations of L-ara, D-ara, or sucrose, irrespective of whether the flies have problem detecting it (as in the case or D-arabinose or L-arabinose) or no problem (as in the case of sucrose) flies don’t form any long-term memory. Therefore there appears to be no obvious relationship between how well a sugar is detected and formation of long-term memory. It is possible that a certain amount of sugar is needed to generate the appropriate signal; however we have no experimental evidence to support such a possibility.

*6a) The observation that two structurally related sugars can evoke different temporal patterns of GCaMP activation in the same neuron is novel, but needs to be explored further. With the GCaMP analysis provided, the differences in activation patterns evoked by D-ara and L-ara in Gr43a neurons are not entirely convincing and have not been investigated to the extent that is necessary. In particular, the authors should examine these patterns across concentrations, and quantify the temporal differences in activation patterns rather than presenting only peak values (which are not different). Ideally, tip-recordings would provide higher resolution and more convincing data to determine if L-ara and D-ara are eliciting distinct neural patterns that could allow flies to discriminate between the 2 molecules. In this scenario, dose responses of the two sugars should be tested with tip recordings of proboscis taste sensilla (or leg taste sensilla).*

To more finely determine whether individual sensory neurons’ activation differs in response to D- and L-arabinose, we have collaborated with Frederic Marion-Poll and his group at AgroParisTech. Their recordings unequivocally demonstrate that f5V tarsi neurons (which express Gr43a) respond differentially to L- and D-arabinose. (Figure 3)

*6b) Additionally, it would be useful to know how the response patterns of D-ara and L-ara compare with those of sucrose and fructose, which are canonical activators of these neurons. Also, what is the role of Gr43a in the temporal kinetics of the response (as opposed to the strength of the response)?*

We have looked at the response of Gr43a neurons with fructose and glucose (Figure 3—figure supplement 2). But because we have removed the emphasis on temporal kinetics of the calcium imaging responses, we feel that the role of Gr43a in that temporal response is no longer pertinent.

*7) The conclusion that different patterns of activation within the same neurons, rather than activation of different sets of neurons, may be important for different behavioral outcomes is based on pulsed activation using optogenetics. A major gap that remains is the question of what pulsed activation mean in terms of calcium activity, and how that connects with possible differences in activation patterns evoked by D-ara and L-ara. Also, pulsed activation of both Gr5a-GAL4 and Gr43a-GAL4 can form LTM, but previous evidence suggests that they are not expressed in the same subsets of neurons.*

We agree with the reviewers and based on their suggestion we have removed most of the opotogenetic experiments and we don’t claim that pattern of activation is important. We have now included only the data that stimulation with light but not heat can substitute the sugar stimulus. In fact the electrophysiological recording suggest that D-ara indeed produces more spikes/s than L-ara, and therefore that more activation does not correlate with more memory.

However, as the reviewers have suggested, since we don’t know the neuronal activity following different amounts of light, we don’t claim that opotogenetic experiments reflect L- and D- stimulation.

We have also kept the Gr43a and Gr5a activation data since our split-Gal4 labeling and other studies suggest there are some Gr43a neurons that also expresses Gr5a. However, if the reviewers think still there is not enough evidence for the coexpression of these receptors we would be willing to remove the data.

*8) In the same section, "To test the possibility that long-term memory requires activation of sensory neurons in a more temporally refined manner, we used the red-shifted channelrhodopsin variant ReaChR" – it is crucial to know whether pulsed stimulation gives rise to the same number of spikes than a (chopped) continuous pulse. From the study of Inagaki et al., it seems that despite the fact that ReaChR triggers more tonic responses, it is still quite phasic. A pulsed stimulus could be more efficient in terms of total number of action potentials generated. In other words, this experiment is not conclusive regarding the frequency of the stimulation, as it can be read also that patterned stimuli are more efficient simply because more spikes are generated. This assumption is wrong (same paragraph): "For example, certain pulsated patterns of light (20 Hz, 15 msec pulse-width for 2.5 ms, repeated every 20 seconds) generated long-term associative memories, while the same amount of light delivered in a uniform, tonic pattern was markedly less effective (15 msec pulse-width repeated every 327 msec continuously) (Figure 4)." – it could be less effective because less spikes are generated. The "amount of light" is not a trustworthy indication of how much spiking is generated in the neurons.*

As we have indicated above, based on reviewers comments we have reduced the amount of optogenetic data, and look forward in the future to examining more closely the relationship between optogenetic activation and electrophysiological responses.

[Editors' note: the author responses to the re-review follow.]

*Essential revision*

*The authors should clearly acknowledge that the mechanism by which sensory neurons communicates information on L-Ara quality and concentration as relevant to LTM remains unanswered. This important ambiguity should be clearly stated in all relevant parts of the text and acknowledged by addition of a single line in the revised Abstract.*

We agree with the reviewer’s assessment and we have included the following statements in the Abstract, Introduction and Results.

Abstract- “However, how sensory neurons communicate information about L-arabinose quality and concentration—features relevant for long-term memory—remains unknown.”

Introduction- “However, the exact mechanism by which these sensory neurons communicate the relevant features of L-arabinose to higher-order systems remains unclear at this stage.”

Results: “However, further work is necessary to determine exactly which subsets of neurons contribute to L-arabinose memory, and whether these neurons needs to be activated in a specific pattern to elicit long-term memory.”

*Recommended revisions (not essential, but to be considered seriously).*

*1) The following is a suggestion for a presentation that may be easier for the non-expert reader.*

*The current organisation of the almost overwhelming amount of data make it very difficult to read the manuscript and the reader is often disoriented by having to go back and forth between different figures. The reviewers suggest a reorganization of the figures in the following order.*

*Figure 1: Immediate preference for a sugar is not predicative of long-term memory formation – different sugars from fruits. Figure 2: Immediate preference for a sugar is not predicative of long-term memory formation – L- vs D-arabinose. Figure 3: The roles of Gr5a, Gr43a and Gr61a expressing neurons in immediate preference and memory formation. Figure 4: Peripheral Gr43a expressing neurons in the foreleg are critical for memory formation. Figure 5: Evoked-calcium activity and spike response of the f5v neuron in response to L- and D-arabinose. Figure 6: Activation of the Gr43a neurons mimics sugar stimulation in memory formation.*

We are grateful for this suggestion and we have organized the figures and the text as suggested.

*2) The interpretation of some of observations from the receptor mutants is difficult, because a few of are imprecise mutations. It may be best to only retain Gr43a and Gr61a mutant data, which make the authors' points. The essential data crucial for the paper is about which neurons mediate memory formation. For this reason, the GAL4-mediating silencing experiments are important and the Gr43a and Gr61a mutations augment the argument. The rest are potentially distracting. This point applies to the current Figure 2 and all supplemental information.*

Based on reviewers suggestions we have simplified the Gr mutant data and only included the Gr43a and Gr61a data in the main figure as well as figure supplement.

*3) For most of the figures, ANOVA followed by the Tukey's test was used. Typically, statistical significance was denoted by using different letters such as a, b, and c.*

We have reorganized the figures and indicated statistical significance using letters as suggested by the reviewer.

*4) Figure 1—figure supplement 2. For panel C, few people have the working memory to figure out the preference order of the different sugars. Using a matrix plot of three columns (D-arabinose, L-arabinose, and L-sorbose) and three rows (L-fucose, D-arabinose, and L-arabinose) for the preference comparison would provide a better illustration. Furthermore, the established order should be maintained in the panels D and F. A direct plot of memory index against immediate preference would be even better.*

We have organized the preference and memory plots in a manner that is easier to follow. We have also presented the data using a matrix plot in the figure supplement for easy and quick comparisons of the mean values of the preference, short-term and long-term memory.

*5) Figure 1—figure supplement 3. The panel order is not matched with that in the legend.*

We have corrected this in the revised version.

*6) Figure 3—figure supplement 1. The inconsistence with the published paper (Fujii et al., 2015) needs to be mentioned. Figure 2 in that paper indicates that the f5v neuron expresses the Gr43a receptor but not Gr5a. However, here the authors found that the f5v neuro expresses both Grs.*

We have included the following statement in the results “Previous studies suggest that in the f5 neurons in the distal tarsi, Gr43a is coexpressed with Gr61a (Figure 4—figure supplement 1) (Freeman and Dahanukar, 2015) There were uncertainities about the coexpression of Gr43a and Gr5a in distal tarsi. However, split-GAL4 reconstitituion assay suggest that Gr5a and Gr43a are likely to be coexpressed in the f5 neurons in the distal tarsi, in agreement with previous work (Miyamoto et al., 2012).”

*7) Introduction. Confusing references. "A specific illustration of this phenomenon is seen with the two chemically similar sugars, D- and L-arabinose: flies greatly prefer D-arabinose to L, but form long-term memories of L-arabinose (a sugar released from pectin as fruit ripens) more consistently (Ahmed and Labavitch, 1980, Dick and Labavitch, 1989)". The references cited discuss L-arabinose being a sugar in ripened fruits and have nothing to do memory formation.*

The reviewer’s point is well taken. We have removed the reference.

*8) It would be helpful to have a graphical summary of the different Gr-GAL4 lines in the various neurons.*

Based on the reviewer’s suggestion we have clarified our observations and explained some ambiguity in clearly identifying the sensilia type. However, to provide a graphical summary of Gr-expression pattern we have to use some of the published data and in the literature there are some uncertainties about the receptor localizations. Since we have not exhaustively analyzed which of these published studies are correct using independent methods and it is not the central issue of our study, we feel we may unintentionally step into a debate that we are not helping to resolve. Therefore, if the reviewers agree, we would rather not summarize works of others in a manner that may be misrepresentation of their work. We hope the field would draw conclusion from our experimental observations.